# An ancient bacterial zinc acquisition system identified from a cyanobacterial exoproteome

**Cristina Sarasa-Buisan**[1☉¤], **Jesús A. G. Ochoa de Alda**[2☉], **Cristina Velázquez-Suárez**[3‡],
**Miguel Ángel Rubio**[3‡], **Guadalupe Gómez-Baena**[4], **María F. Fillat**[1]*, **Ignacio Luque**[3]*

**1** Departamento de Bioquímica y Biología Molecular y Celular e Instituto de Biocomputación y Física de Sistemas Complejos (Bifi), Universidad de Zaragoza, Zaragoza, Spain, **2** Didáctica de las Ciencias Experimentales y la Matemáticas, Universidad de Extremadura, Cáceres, Spain, **3** Instituto de Bioquímica Vegetal y Fotosíntesis, CSIC and Universidad de Sevilla, Seville, Spain, **4** Departamento de Bioquímica y Biología Molecular, Universidad de Córdoba, Córdoba, Spain

☉ These authors contributed equally to this work.
¤ Current address: Instituto de Bioquímica Vegetal y Fotosíntesis, CSIC and Universidad de Sevilla, Seville, Spain
‡ CV-S and MAR also contributed equally to this work.
* fillat@unizar.es (MFF); ignacio.luque@ibvf.csic.es (IL)

## Abstract

Bacteria have developed fine-tuned responses to cope with potential zinc limitation. The Zur protein is a key player in coordinating this response in most species. Comparative proteomics conducted on the cyanobacterium *Anabaena* highlighted the more abundant proteins in a *zur* mutant compared to the wild type. Experimental evidence showed that the exoprotein ZepA mediates zinc uptake. Genomic context of the *zepA* gene and protein structure prediction provided additional insights on the regulation and putative function of ZepA homologs. Phylogenetic analysis suggests that ZepA represents a primordial system for zinc acquisition that has been conserved for billions of years in a handful of species from distant bacterial lineages. Furthermore, these results show that Zur may have been one of the first regulators of the FUR family to evolve, consistent with the scarcity of zinc in the ecosystems of the Archean eon.

## Introduction

Correct metallation of proteins relies on maintaining balanced concentrations of metals in the cytoplasm, based on the operation of complex mechanisms to cope with the large fluctuations in metal availability observed in nature [1]. Acclimation mechanisms maintain balanced concentrations by fostering the acquisition of limiting metals and detoxifying those in excess [2]. Zinc is an essential metal for life, being present in approximately 5% to 10% of all proteins, where it plays structural, catalytic, or regulatory roles [3]. Bacteria are often exposed to changes in zinc availability. This metal is particularly limiting in the open ocean, in habitats with strong competition for nutrients (i.e., microbial mats), or in infected tissues, where the host actively sequesters zinc to block the proliferation of pathogens. Strategies for acclimation to zinc limitation are diverse. They include the operation of sensing mechanisms, the induction of high-

**Data Availability Statement:** Data obtained by mass spectrometry was deposited in the PRIDE

repository under the accession number PXD036795.

**Funding:** This work was funded by grant 438 PID2019-104889GB-I00 from Ministerio de Ciencia, Innovación y Universidades (https://www.ciencia.gob.es) awarded to MFF; by grant E35_20R Biología Estructural, Gobierno de Aragón (https://www.aragon.es/organismos/departamento-de-educacion-ciencia-y-universidades/direccion-general-de-ciencia-e-investigacion) awarded to MFF and by grant PID2021-128477NB-I00, Ministerio de Ciencia e Innovación /Agencia Estatal de Investigación/10.13039/501100011033/ FEDER, UE (https://www.ciencia.gob.es) awarded to IL. The funders had no role in study design, data collection and analysis, decision to publish, or preparation of the manuscript.

**Competing interests:** The authors have declared that no competing interests exist.

**Abbreviations:** EMSA, electrophoresis mobility shift assay; DDA, data-dependent acquisition; DIA, data-independent acquisition; FDR, false discovery rate; GI, Growth Index; GOE, Great Oxygenation Event; HGT, horizontal gene transfer; LBCA, last bacterial common ancestor; MSA, multiple sequence alignment; OM, outer membrane; ROS, reactive oxygen species; TBDT, TonB-dependent transporter; TSS, transcription start site; ZBS, Zur-binding site.

affinity transport systems, the mobilization of zinc ions from cell reservoirs, the replacement of zinc metalloproteins by zinc-independent paralogs, or the shift of metabolic flow toward pathways that do not involve zinc-containing proteins [2,4,5].

Zinc sensing in bacteria is exerted by regulators of the MerR, ArsR/SmtB, TetR, MarR, and the FUR family, which perceive either deficiency or sufficiency. Zur is a widespread member of the FUR family that senses zinc. Other members of this family sense iron (Fur), nickel (Nur), manganese (Mur), hydrogen peroxide (PerR), or haem (Irr) [6,7]. Proteins of the FUR family are dimeric, each monomer composed of an N-terminal winged-helix domain for DNA binding and a C-terminal dimerization domain connected by a mobile hinge. Zur senses zinc availability through a coordination site in the hinge between both domains [8–11]. The affinity of this site for zinc is commonly in the sub-picomolar range, matching the regular concentration of free zinc in the cytoplasm [12]. Zinc binding to this regulatory coordination site promotes a structural change toward an optimal conformation for binding to DNA [8–11,13]. Hence, when zinc is sufficient, Zur binds to DNA and acts in most cases as a repressor, as Zur binding sites overlap with the occupancy site of RNA polymerase in the promoter of regulated genes. Zinc concentration dropping below a threshold provokes the induction of the Zur regulon. The characterization of the Zur regulon in various bacteria has brought into evidence that genes repressed by Zur are generally involved in optimizing zinc handling and maximizing zinc uptake under scarcity (see [5] and references therein).

Interestingly, in some bacteria Zur controls the expression of biosynthetic clusters for the synthesis of zincophores, low molecular weight organic molecules secreted to the extracellular milieu to scavenge trace zinc. Once metallated, zincophores are presumably imported through the outer membrane by TonB-dependent transporters (TBDTs) [14–16]. This illustrates that zinc handling may involve components within the cell and organic molecules of the extracellular milieu under scarcity. The external medium also contains proteins that constitute the so-called exoproteome. The importance of the exoproteome has been long neglected, but in recent years its relevance for a variety of functions, including sinking and predator avoidance, biofilm formation, extracellular substrate degradation, virulence or intercellular interaction, and competence has been demonstrated [17–21].

Cyanobacteria are gram-negative photosynthetic bacteria that contain a large amount of metalloproteins for light harvesting, photosynthetic electron transport, redox metabolism, carbon or nitrogen fixation, and other functions [22,23]. Zur has been well characterized in a few species. Remarkably, in the oceanic cyanobacterium *Synechococcus* sp. WH8102, Zur exhibits unique features regarding the orientation of the 2 domains and the regulatory site, which is distinct and involves coordination residues different from those in non-cyanobacterial Zur [24]. These features may be general for other cyanobacterial Zur proteins. In the multicellular filamentous cyanobacterium *Anabaena* sp. PCC 7120 (also known as *Nostoc* sp. PCC 7120), the Zur regulon has been shown to include a high-affinity transport system for zinc, TBDTs of the outer membrane, putative metallochaperones of the COG0523 family, paralogs of zinc-containing proteins, and proteins involved in detoxification and protection from reactive oxygen species (ROS) [25–27]. Some of these Zur targets were essential for acclimation to low zinc conditions [28]. Besides, Zur (also known as FurB in this strain) was shown to regulate *furA*, encoding the iron uptake regulator, suggesting a possible crosstalk between iron and zinc-responsive regulons to fine-tune metal homeostasis [26].

Zur targets in *Anabaena* include proteins of the cytoplasm, plasma membrane, periplasm, and outer membrane. However, little is known about the possible contribution of the exoproteome to acclimation to low zinc conditions or about the dynamics of the exoproteome upon changes in zinc availability. In this work, we present a quantitative analysis of the exoproteome of wild-type *Anabaena* and a *zur* mutant. Results from our analyses indicate a relevant role of

the exoproteome in the acclimation to zinc limitation, expanding the knowledge of the components involved in this phenomenon. While the precise role of each exoprotein in this response is to be determined, here we show evidence indicating that a thus far uncharacterized protein, named here as ZepA, is a zinc-binding protein highly enriched in the Δ*zur* mutant exoproteome that mediates zinc supply to *Anabaena* cells under normal and under zinc limitation conditions. Moreover, evidence indicates that this protein is a descendant of a very ancient system for zinc acquisition that likely operated already in the last bacterial common ancestor (LBCA).

## Results

### The exoproteome of *Anabaena* sp. PCC 7120 and the Δ*zur* mutant

To characterize the exoproteome of *Anabaena*, cells were grown under standard conditions until reaching the late exponential phase and the extracellular milieu was processed and analyzed as described in the Materials and methods section. Then, a comparative study of the exoproteomes of the Δ*zur* mutant versus wild-type *Anabaena* was performed using a quantitative, data-independent acquisition (DIA) [sequential windowed acquisition of all theoretical mass spectra (SWATH-MS)] approach. Two biological replicates from each strain, each analyzed in triplicate, were used in the present study. Pearson's correlation coefficients indicated good reproducibility between biological replicates ($R^2 = 0.960$ for the wild type and $R^2 = 0.789$ for the Δ*zur* replicates, S1 Fig). A total of 347 proteins were identified and quantified as components of the exoproteome of *Anabaena* sp. PCC 7120 and the Δ*zur* mutant.

First, we compared the set of proteins identified here to previously reported exoproteomes of *Anabaena* under similar growth conditions [29,30]. We found that 69% of the proteins identified by Oliveira and colleagues and 73% of the proteins in the Hahn exoproteome were also identified in our study. However, we were able to identify 285 proteins previously unknown to be components of the *Anabaena* exoproteome (S1 Table). Computer-based predictions of the subcellular localization and potential secretion pathways of the proteins identified in our study predicted over 65% of these to be secreted or to have a peripheral localization. Out of the 347 identified proteins, 131 contained putative N-terminal signal peptides for the Sec or the Tat translocases, nominating them as targets for insertion or translocation through the plasma membrane [31]; 98 did not contain a putative signal peptide but were predicted to be secreted or to have a peripheral localization (i.e., plasma membrane, periplasm, or outer membrane); 98 proteins were predicted to be cytoplasmic, and 20 were of unknown localization (Fig 1A and S2 Table). These analyses as a whole indicate that the group of proteins identified is primarily enriched in proteins that are likely secreted or located in peripheral compartments of the cell. This enrichment is notable given that only 9% of the proteins encoded in the *Anabaena* genome (549 out of 6,070 CDS) contain a signal peptide [32], suggesting that the proteins detected here represent a distinct subset of the organism's proteome.

A quantitative comparison of the exoproteomes of the Δ*zur* mutant and the wild-type strain revealed compositional differences, 10 proteins showing significant differences in abundance between the 2 strains (false discovery rate (FDR) < 0.05, $\log_2$ fold change > 2) (Fig 1B and S2 Table). Six proteins were overrepresented (hereafter O-proteins) and 4 were underrepresented (hereafter U-proteins) in the exoproteome of the Δ*zur* mutant with respect to the wild type (S3 Table and Fig 1B, green and red dots, respectively). Support for this analysis was obtained by western blot using antibodies developed against one of the O-proteins, All3515. This protein was detected in the extracellular medium of cultures of the Δ*zur* strain but not of the wild type, which is consistent with the proteomic data. Importantly, All3515 was not

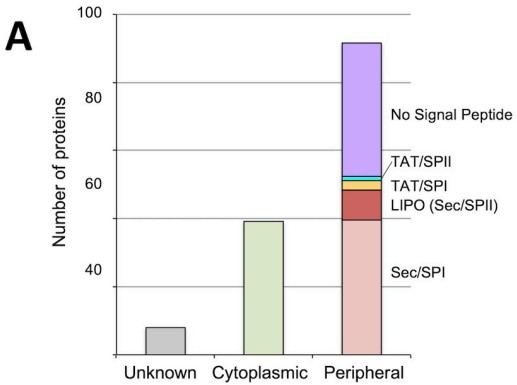

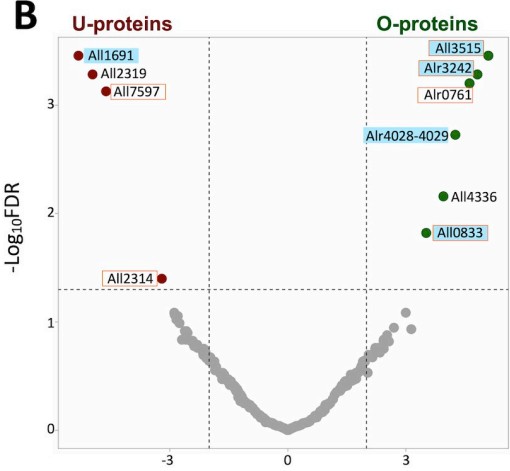

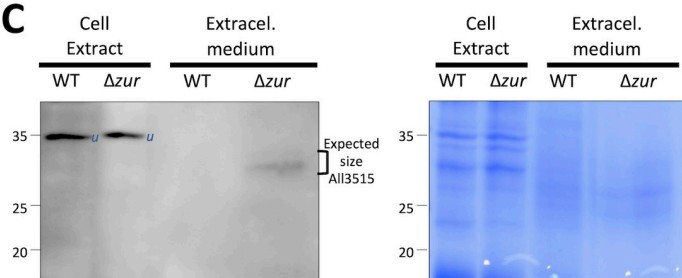

**Fig 1. Composition and differences in the exoproteome of wild-type *Anabaena* and the Δ*zur* mutant.** (A) Histogram displaying the predicted subcellular localization of the proteins identified in the exoproteomes of wild-type *Anabaena* and the Δ*zur* mutant. The presence and type of signal peptide for proteins with peripheral localization is indicated. (B) Volcano plot displaying proteins that show differential abundance in the exoproteomes of the wild type and the Δ*zur* strain. The fold change is defined as the abundance ratio in the Δ*zur* mutant with respect to the wild type (mutant/WT). The data underlying Fig 1A and 1B can be found in S1 Data. Proteins underrepresented (U-proteins) and overrepresented (O-proteins) in the exoproteome of the mutant are shown in red and green color, respectively. A cyan background labels Zur-regulated proteins (see S3 Table) and an orange frame labels proteins containing signal peptides (see S2 Table). (C) Subcellular distribution of the All3515 protein; 50 μg of protein of cell extracts from wild-type *Anabaena* and the Δ*zur* mutant and a volume of a concentrated preparation corresponding to 6 ml of extracellular medium from cultures of those strains were resolved by SDS-PAGE and subjected to western blot with All3515-specific antibodies (left panel) or to Coomassie staining (right panel). Numbers at the left side indicate the MW (in KDa) and a bracket indicate the expected MW of All3515. A band reacting nonspecifically with the anti-ZepA antibody is labeled as "*u*." This experiment was repeated 3 times and similar results were obtained. FDR, false discovery rate.

detected in cell extracts of Δ*zur*, indicating that the presence of this protein in the extracellular milieu is unlikely due to cell lysis (Fig 1C).

O- and U-proteins showed distinct functional profiles. Among O-proteins, Alr3242 and Alr4028-4029 are TBDTs of the outer membrane, All0833 is the periplasmic subunit (ZnuA) of the zinc-specific ZnuABC transporter, All3515 is a protein of unknown function, All4336 is the S10 ribosomal protein, and Alr0761 a hypothetical protein. Among U-proteins, All1691 is the FurA ferric uptake regulator, All2319 is the PII protein that regulates carbon/nitrogen metabolism, All7597 is a hypothetical protein with a domain of unknown function (DUF2808), and All2314 is a putative endoU-like nuclease.

Enrichment of O-proteins in the exoproteome of the mutant suggested that they may be specifically expressed in this strain, i.e., by release from Zur repression. Computational search identified putative Zur-binding sites (ZBSs) in the upstream region of 4 genes encoding O-proteins (S3 Table), which was consistent with previous observations [25]. In genes with mapped transcription start sites (TSSs) [35], we observed that ZBSs overlapped with promoter sequences, which is compatible with a repressor role for Zur. FIMO search also identified 2 putative ZBSs upstream of one U-protein gene, *all1691* (S3 Table). In this case, the ZBSs were considerably far upstream (51 and 115 pb) of the TSS.

## Regulation of *all3515*

In general, information on the function of exoproteome components is scarce for all bacterial species. To fill this gap, we focused on the most overrepresented protein in the exoproteome of the Δ*zur* mutant, All3515. The *all3515* gene was observed to be highly induced in the Δ*zur* mutant [25]. A computational search showed the existence of 2 overlapping putative ZBSs proximal to 2 close TSSs in the promoter region of *all3515* (S3 Table), which is consistent with previous observations, though evidence on direct regulation by Zur was lacking. The interaction of Zur with the upstream region of *all3515* was tested by electrophoresis mobility shift assays (EMSAs). As shown in the right panel of Fig 2A, Zur could bind to a DNA fragment encompassing the 2 overlapping ZBSs (Fig 2E) [33]. The *all3515* fragment was deduced to bind Zur with high affinity by comparison with a positive control previously characterized ($K_d$ = 2.5 nM) (Fig 2A, central panel), which is consistent with the high similarity of both ZBSs to the consensus binding sequence (Fig 2F) [25]. The observation of a single retarded band in EMSA assays indicated that either Zur binds to only one of the 2 sites at a time, simultaneous binding being not possible, or that Zur binds to both sites with high cooperativity, as described for *E. coli* Zur binding to the *znuABC* promoter [8]. Both ZBSs overlap the putative −10 boxes for TSS1 and TSS2, which is compatible with a repressor role, as the binding of Zur would block polymerase access to both promoters (Fig 2E).

Expression of *all3515* was analyzed by northern assays in *Anabaena* cultures grown in the absence or presence of a metal chelator (TPEN) for 24 h. A ca. 1.1 Kb band was observed to be highly induced in cultures treated with TPEN, becoming more intense at late time points, but was absent in cultures under normal conditions (Fig 2B, black arrowhead). The size of this transcript nicely fits with the length of mRNAs starting at TSS1 or TSS2 and encompassing the *all3515* ORF (915 bp) (Fig 2D and 2E). A longer band of ca. 2.1 Kb was visible both in cultures in normal conditions and cultures treated with TPEN (Fig 2B, white arrowhead). This transcript abundance did not change in any condition and its size is compatible with transcription initiation at the TSS4 located within the upstream *all3516* gene (Fig 2D and 2E). Northern assays also showed a high abundance of the 1.1 Kb transcript in the Δ*zur* mutant in the presence or absence of TPEN (Fig 2C), consistently with Zur regulating TSS1 and TSS2 by repression. Notice that, the reduced level of both transcripts in the culture with TPEN is probably a

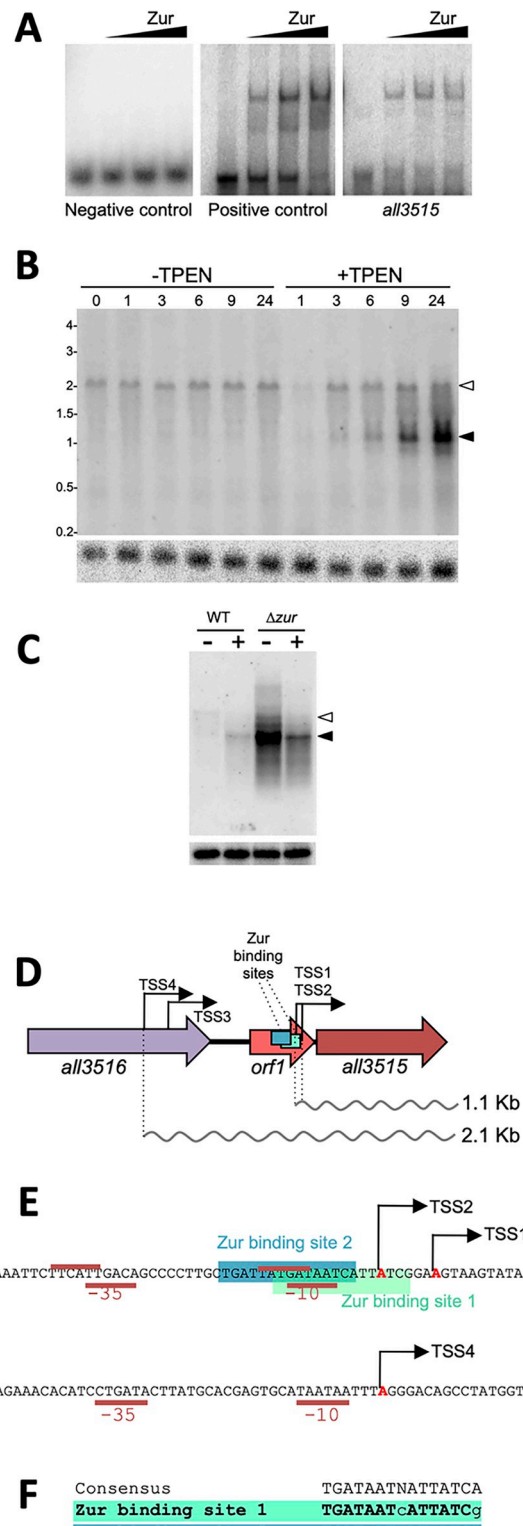

**Fig 2. Regulation of *all3515* expression.** (A) Zur binding to the upstream region of *all3515* was tested by EMSA with increasing concentrations of Zur protein (0, 1, 5, 10 pmol in lanes from left to right). A parallel control assays were performed with the upstream region of *all4725*, previously shown to bind Zur with high affinity and with a mutated version of this fragment as a negative control. This experiment was repeated twice with similar results. (B) Northern assay using RNA from *Anabaena* cells cultured in the absence or presence of 20 μm TPEN for the indicated length of

time. Solid and empty arrowheads indicate respectively the 1.1 kb and 2.1 transcripts hybridizing with an *all3515* probe. The bottom panel shows the same membrane after hybridization with a probe of the *rnpB* gene for normalization. This experiment was repeated twice and similar results were obtained. (C) Northern assay with RNA from *Anabaena* or Δ*zur* cells cultured in the absence or presence of 20 μM TPEN for 24 h. Details are like in B. (D) Structure of the genomic region of *all3515* showing the position of the TSSs mapped by Mitschke and colleagues. (E) Sequence of the promoter regions of *all3515* indicating the ZBSs and the −10 and −35 boxes. (F) Comparison of the ZBSs in the *all3515* promoter with the consensus reported by Napolitano and colleagues. Conserved nucleotides are shown in bold capital letters. EMSA, electrophoresis mobility shift assay; TSS, transcription start site; ZBS, Zur-binding site.

consequence of the reduced activity of RNA polymerase, a zinc-containing enzyme. Overall, these results evidenced that *all3515* is expressed from 2 proximal promoters controlled by Zur and from a distal promoter that provides a constitutive basal level of expression irrespective of the zinc regime (Fig 2D). This transcription pattern is consistent with the levels of the All3515 protein in the extracellular space (Fig 1C).

## Functional characterization of All3515

All3515 is a protein of unknown function with no homologs characterized to date. To get an insight into its function, insertion-deletion mutants were generated by gene replacement in *Anabaena* and were subjected to phenotypic characterization (Figs 3 and S2C). Two independent *all3515* mutants were tested in parallel to the wild type. As shown in Fig 3A, the growth of the *all3515* mutants was similar to that of the wild type in normal conditions (left panels), but mutants were impaired for growth in the presence of metal chelators like EDTA or TPEN, indicating a reduced capacity for these mutants to thrive in media depleted for metals. These results, together with its presence in the exoproteome hinted at a possible relation of All3515 with the extra-cytoplasmic handling of metals. Growth assays were carried out to test this in the presence of increasing concentrations of different metals. As shown in Fig 3B, *all3515* mutants grew better than the wild type in the presence of high concentrations of zinc but showed some impairment for growth on plates with high concentrations of cadmium, nickel, or copper. Quantification and statistical analysis of the results shown in Fig 3A and 3B is shown in S10 Fig. Combined, these results would be consistent with a role of All3515 in the provision of zinc to the cell (see S3 Fig). For example, the enhanced tolerance of the mutants to high zinc suggests that in the absence of All3515, this metal does not reach high levels in the cytoplasm, precluding deleterious mismetallation of proteins or oxidative stress. Conversely, impaired growth of the mutants in the presence of chelators could be interpreted as a consequence of an insufficient supply of zinc, as if the absence of All3515 exacerbates the deficiency of this metal imposed by the chelators.

It is interesting to point out that the reduced susceptibility of the mutants to high zinc concentrations (Figs 3B and S10) indicates that in the wild type, All3515 is active in these conditions, which is in agreement with the constitutive expression observed for *all3515* (2.1 Kb transcript in Fig 2B). Likewise, the growth advantage of the wild type in the presence of chelators (Figs 3A and S10) indicates the operation of All3515 under low zinc, which is consistent with the transcriptional de-repression observed (Fig 2B and 2C). The enhanced expression of *all3515* under low zinc appears in line with a mechanism to maximize zinc supply by means of increasing the total number of All3515 molecules (S3 Fig).

According to this interpretation, *all3515* mutants would be expected to show a permanent impairment in the supply of zinc, which may be fatal under zinc limitation (Figs 3A and S10) but protective in elevated zinc concentrations (Figs 3B and S10). In such a scenario of chronic zinc deficiency, it is not surprising that high concentrations of other metals provoke toxicity to

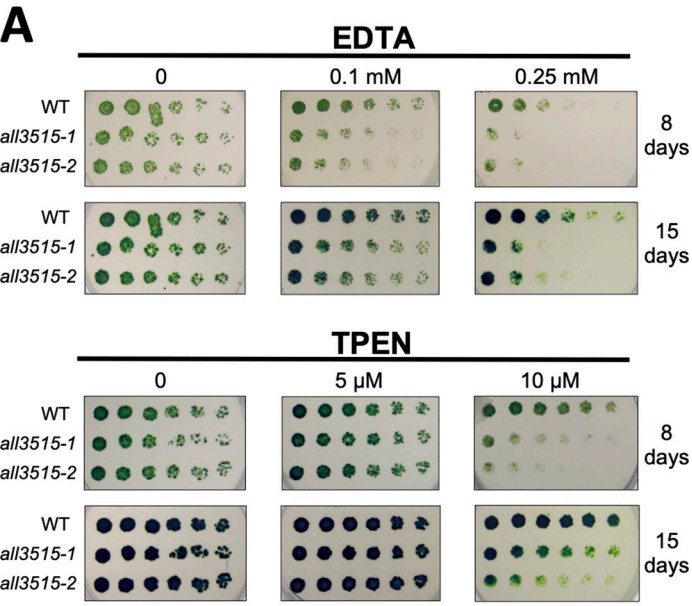

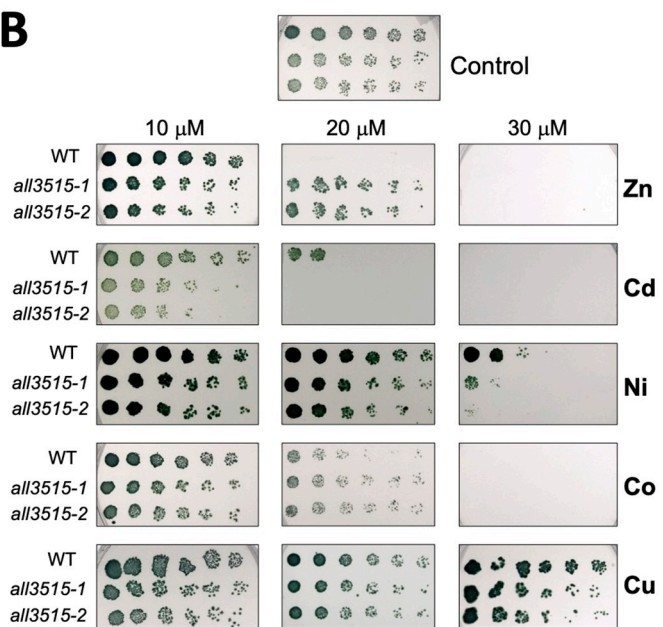

**Fig 3. Growth of wild-type *Anabaena* and Δ*all3515* mutants in the presence of chelators or high concentrations of metals.** (A) Growth of wild-type *Anabaena* and 2 independent clones of the Δ*all3515* mutant was assayed for the indicated length of time on BG11 plates supplemented with the indicated concentrations of EDTA or TPEN. (B) Wild-type *Anabaena* and 2 independent clones of the Δ*all3515* mutant were assayed for growth on BG11 plates supplemented with the indicated concentrations of metal salts (ZnSO₄, Cd acetate, NiSO₄, CoCl₂, or CuSO₄) shown on the right side. Quantification and statistical analysis of experiments in Fig 3A and 3B is shown in S10 Fig.

the mutants due to mismetallation of zinc sites, which is consistent with the observation of impaired growth of the mutants in media with excess Cd, Ni, or Cu (Figs 3B and S10). We propose the name of ZepA (zinc-handling extracellular protein A) for All3515.

## Biochemical characterization of ZepA

The identification of ZepA in the exoproteome of *Anabaena* (Fig 1B and 1C) is consistent with the presence of an N-terminal signal peptide for the Sec system and a C-terminal PEP-CTERM domain in its sequence (Figs 4A and S2A). The latter is a ca. 25 amino acids domain exclusive of gram-negative bacteria that displays an invariable Pro-Glu-Pro motif, a stretch of hydrophobic amino acids and a cluster of basic residues (S2B Fig). N- and C-terminal sequences for exportation are expected to be cleaved upon passage through the plasma membrane [31,34]. Evidence of the cleavage of ZepA upon exportation was obtained in a heterologous host. *Anabaena* ZepA was found to be highly toxic in some *E. coli* strains, but full-length ZepA could be expressed using a tightly controlled system in the Lemo21(DE3) strain [35], which revealed to some extent, tolerance to this protein. Distinct bands between 25 and 35 KDa reacting to a ZepA-specific antibody were observed in different fractions of these cells. Out of the 3 reactive bands (*a*, *b*, and *c* in Fig 4B), the smallest one (band *c*) was detected only in the periplasm and the extracellular medium. These observations indicate that ZepA is also secreted in this host and are compatible with a double cleavage of ZepA upon secretion. Moreover, the size of the smallest band in the extracellular medium of Lemo21(DE3) is similar to the size of ZepA in the extracellular medium of the Δ*zur* mutant of *Anabaena* (S2D Fig), suggesting cleavage at similar positions in both organisms.

The proposed involvement of ZepA in zinc supply to the cell (Fig 3) suggested a direct interaction of the protein with this metal. The predicted mature sequence of ZepA was fused to a C-terminal Strep-tagII and overexpressed to a high level in the periplasm of Lemo21(DE3) following standard procedures [36]. Recombinant ZepA-StrepTag-II was purified by affinity chromatography and showed an apparent molecular weight of 27,460 Da in gel filtration experiments (Fig 4C and 4D), in good agreement with the expected size of this protein as predicted from its sequence (27,384 Da). This size was also consistent with its migration in SDS-PAGE and fits with a monomeric conformation for ZepA. Pure preparations of recombinant ZepA were subjected to preparative gel filtration and fractions were analyzed by ICP-OES. As shown in Fig 4E, zinc, but not other metals co-eluted with the protein in the chromatography. Fractions from 2 biological replicates revealed that the ZepA monomers binds 1.48 (±0.15) zinc atoms on average, significantly above a 1:1 ratio (one-tailed Student's $t$ test, $p$-value = 0.07). This finding suggests the presence of a high-affinity zinc-binding site on the ZepA monomer, accompanied by a secondary, lower-affinity site.

## Structural modeling of ZepA

A ZepA model is available in the UniProt database (Q8YRD3), but such model did not considered maturation by cleavage at both ends. As cleavage would occur in an unfolded state, it is unlikely that ZepA would exist as a full-length folded precursor. This prompted us to model the structure of the putative mature protein with Alphafold2, which yielded a model of high confidence (pLDDT 93.5 and ptm score 0.905) (S4 Fig). The model displays 2 well-packed domains interconnected by 3 loops (Fig 4F). Domain A shows an immunoglobulin-like beta sandwich fold formed by 8 beta ribbons, whereas domain B is formed by a 4-ribbon beta sheet, 2 short alpha helices and a loop containing 2 short beta ribbons that extrudes from the structure. This model was compared to structure databases using the DALI server [37]. Domain A showed structural similarity to beta sandwich domains of many proteins, the majority of them located on external layers of prokaryotic and eukaryotic cells. No structure similar to domain B was retrieved, indicating that this domain is the only member of a new fold family.

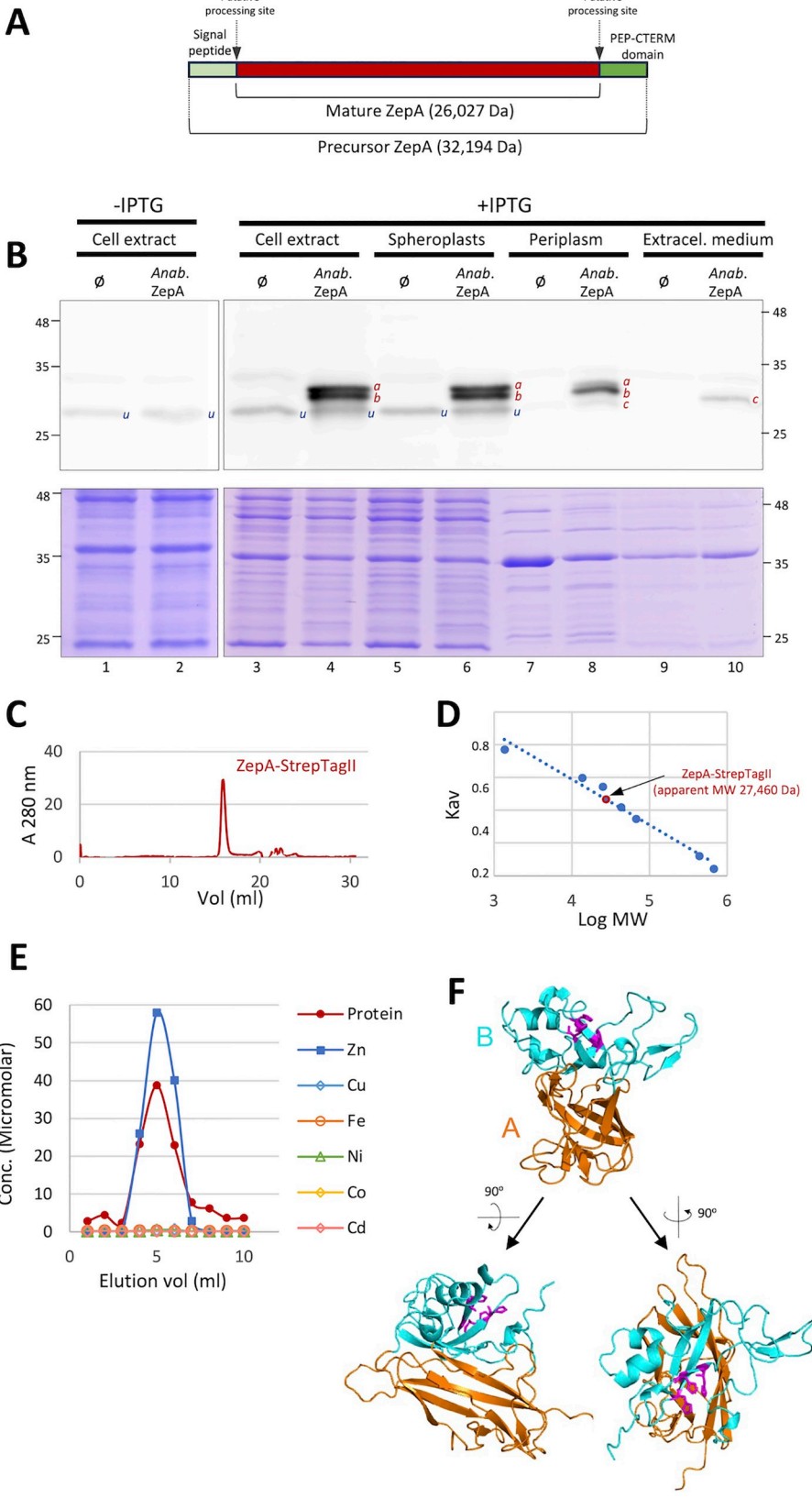

**Fig 4. Biochemical characterization of ZepA.** (A) Schematic representation of ZepA showing the position of the signaling sequences for exportation and putative processing sites. (B) Western blot of cell fractions of *E. coli* Lemo21 (DE3). Lanes labeled as "ø" were loaded with fractions of cells containing the pET28b empty vector, lanes labeled as "*Anab*. ZepA" were loaded with fractions from cells containing plasmid pET28b:ZepA_Ana, which in the presence of IPTG drives expression of full-length *Anabaena* ZepA. Lanes 1 and 2 were loaded with a cell extract corresponding to 50 μl of culture, lanes 3–6 were loaded with cell extract corresponding to 40 μl of culture, lanes 7 and 8 were loaded with periplasmic fraction corresponding to 500 μl of culture, lanes 9 and 10 were loaded with extracellular material corresponding to 1.25 ml of culture. The gel of the top panels was subjected to western blot with a specific antibody against ZepA. Unspecific bands are labeled as "*u*." Bands reacting specifically to the anti-ZepA antibody are labeled as "*a*," "*b*," or "*c*" according to their size. The bottom panel is an identical gel stained with Coomassie. (C) Gel filtration of recombinant ZepA-StrepTag-II. The plot shows the elution profile of a pure preparation of this protein in a Superdex 200 Increase 10/30 GL column. (D) The plot shows the Kav (calculated as explained in Materials and methods) as a function of the logarithm of the molecular weight of proteins subjected to gel filtration. Proteins used for calibration of the column are in blue and ZepA-StrepTag-II is indicated in red color. (E) The plot shows the elution profile of a preparative PD-10 gel filtration column loaded with a pure preparation of recombinant ZepA-StrepTag-II. The concentration of protein in the different fractions, determined by Bradford assay and the concentration of metals, determined by ICP-OES is indicated. The data underlying Fig 4C, 4D, and 4E can be found in S1 Data. (F) Model structure of ZepA obtained using AlphaFold2. Distinct views of the model structure are shown. The 2 domains are depicted in distinct colors and labeled. Histidine residues proposed to coordinate zinc are shown in purple.

## Conservation of ZepA in bacteria

A screen for homologs was conducted in a database of 16,157 representative genomes, including 15,626 bacterial genomes. ZepA homologs were not encoded in Eukarya or Archaea, but were sporadically encoded in a small fraction (0.3%) of bacterial genomes from three distant phyla, Cyanobacteria, Proteobacteria, and Nitrospirota, and the PVC (Planctomycetota, Verrucomicrobiota, Chlamydiota) superphylum (Fig 5A). These homologs were identified based on their high sequence coverage ($\geq$75%), sequence identity ($\geq$35%), and the presence of shared conserved features, such as a predicted N-terminal signal peptide and a C-terminal PEP-CTERM domain that flanked a mature region with many conserved residues (Fig 5B). Alphafold2 modeling of 15 diverse ZepA homologs (aligned in Fig 5B) returned confident protein structures (pLDDT > 0.9) highly similar to the *Anabaena* ZepA model (r.m.s.d. 0.336-1.369 Å in a minimum of 152 α-C atoms aligned) (S5A and S5B Fig).

## Zinc coordination site(s) in ZepA and ZepA homologs

Potential zinc-binding sites were predicted for *Anabaena* ZepA and homologs using the MIB2 server [38], and 15 to 25 potential coordination residues were identified in each protein, with the top MIB2 scores corresponding to 3 conserved histidines (i.e., His44, 53, and 55 of the mature *Anabaena* ZepA, corresponding to His80, 89, and 91 of the precursor). Despite being part of the less conserved regions of the protein, these residues showed almost absolute conservation and were predicted for each ZepA homolog as zinc-binding residues by MIB2 (Figs 6A, S4A, and S4B; see also an extended version of Fig 6 in S6 Fig). This, together with their close spatial arrangement in the model structures (less than 3.8 Å between N atoms of the histidine side chains in the *Anabaena* ZepA), strongly suggested that they may act as coordination residues for a zinc atom [39] (Figs 6B, S4A, and S4B). In all ZepA homologs, these 3 histidines were found in a motif with the sequence H-X$_{2-9}$-H-F/Y-H, where X is any amino acid (Figs 5B, 6A, and S6). Zinc ions are typically coordinated in a tetrahedral geometry [40]. For ZepA, it is possible that a fourth coordination site could be provided by a water molecule or by a highly conserved glutamate residue (position 82 in the *Anabaena* mature protein and 118 in the precursor) located in a flexible loop that could become closer to the histidine cluster in the holoprotein (S5C Fig). Conservation of the zinc-binding site further supports the involvement of ZepA homologs in zinc handling.

With a lower score, MIB identified zinc-binding residues in the N-termini of mature ZepA proteins, an unstructured tract that in *Anabaena* includes 3 histidines (His1, His4, and His6)

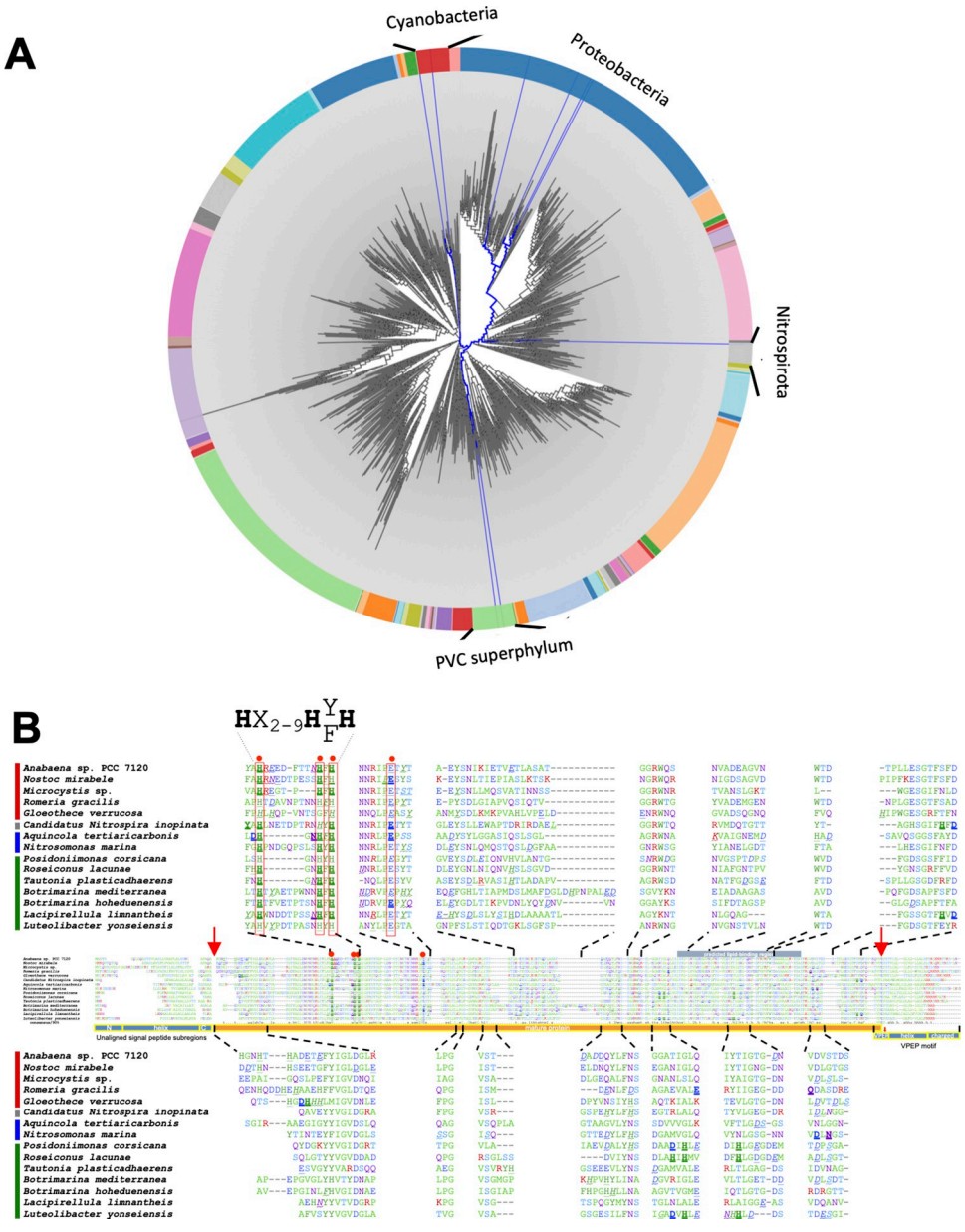

**Fig 5. ZepA is conserved in distant bacterial phyla and has an ancient origin in evolution.** (A) Distribution of ZepA homologs (blue branches) across the evolutionary tree of bacteria (gray). The tree includes all known phyla and each portion of the circle delimits a distinct phylum. Phyla with ZepA homologs are labeled at the outer rim to avoid clutter. The data underlying this figure can be found in S1 Data. (B) Multiple sequence alignment of a representative set of ZepA homologs. Red arrows mark potential cleavage sites of N-terminal signal peptides (according to SIGNALP6.0 prediction) and C-terminal VPEP motif (predicted by analogy to the LPXTG motif of substrate proteins of the *Staphylococcus aureus* sortase A system) and delimit the sequence of the mature proteins (horizontal orange bar). Blue horizontal bars at the N termini label the subregions of signal peptides (the N-terminal, the hydrophobic, and the C-terminal) and at the C termini the subregions of the PEP-CTERM region, including the VPEP motif that includes a highly conserved Proline-Glutamate-Proline sequence, a hydrophobic putative transmembrane region and a terminal positively charged segment. Over and under the alignment, a close view of some regions is shown to illustrate the putative $Zn^{2+}$ binding amino acids identified by MIB2. These amino acids are formatted and underlined according to their MIB score: between 2 and 3 (i.e., *H*), between 3 and 4 (i.e., H), and >4 (i.e., H). Red dots and red frames label putative $Zn^{2+}$ binding residues conserved in more than 80% of sequences. The consensus sequence of the putative zinc binding site is shown at the top.

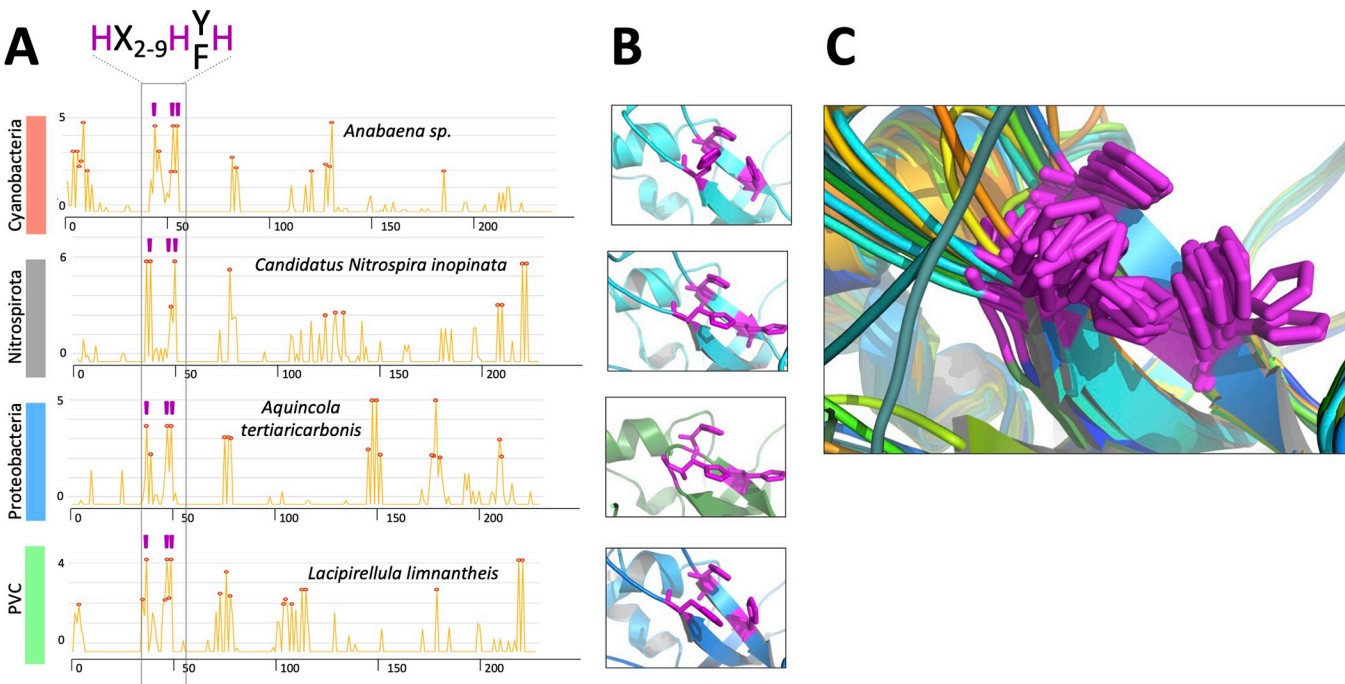

**Fig 6. Prediction of metal-binding residues and conservation of the spatial arrangement of putative zinc-binding histidines in ZepA homologs.** (A) Plots correspond to the output of the MIB2 metal binding prediction server. The *y* axis indicates the MIB2 score and the *x* axis indicates the amino acid positions. The species and the phylum of each ZepA homolog is indicated. Histidines predicted to form the zinc-binding pocket are enclosed in a frame and indicated with a purple bar. The consensus sequence of the putative zinc-binding pocket is shown at the top with histidines residues in purple. The data underlying this figure can be found in S1 Data. (B) Pictures show a close view of the zinc-binding pocket of each ZepA homolog. The structure of each protein was modeled with AlphaFold2. Putative zinc binding histidine residues are depicted in purple color. (C) Superposition of all structures shown in S6B Fig to illustrate the similarity of the putative zinc binding site of all ZepA homologs.

and 3 acidic residues (Asp8, Glu9, and Glu11) (Fig 6B), which is reminiscent of the flexible histidine-rich N-termini, of proteins involved in zinc transport such as ZnuA and ZinT [41,42]. This would be consistent with the observed zinc:ZepA stoichiometry (Fig 4E).

Putative interaction sites for molecules of the cell periphery (i.e., lipopolysaccharide, N-acetyl glucosamine, or a hopanoid-like molecule) were identified by GalaxySite software (https://galaxyproject.org/use/interactomix/#) in *Anabaena* ZepA, and some of these sites were conserved in ZepA homologs (S7 and S8 Figs), further supporting the peripheral localization of ZepA in distinct species.

## Functional characterization of ZepA homologs

Genome neighborhood analysis showed that *zepA* homologs were commonly flanked by genes encoding metalloproteins, most commonly specific of zinc or related to zinc handling, including threonyl- and cysteinyl-tRNA synthetases, the DksA transcription factor, 6-carboxy-tetra-hydropterin synthase, N-acetyl-muramoyl-L-alanine amidase or the transporters ZnuA and ZupT (Fig 7A). The *zepA* genomic context also included genes encoding metallochaperones of the COG0523 family, TBDTs and other proteins with a PEP-CTERM domain. Furthermore, in species of Proteobacteria, Nitrospirota, and the PVC superphylum, *zepA* was proximal to a gene encoding a regulator of the FUR family (Fig 7A). This gene's identity cannot be deduced merely from the sequence, but the enrichment of the neighborhood in genes encoding zinc-binding proteins, strongly supported that this gene likely encodes Zur.

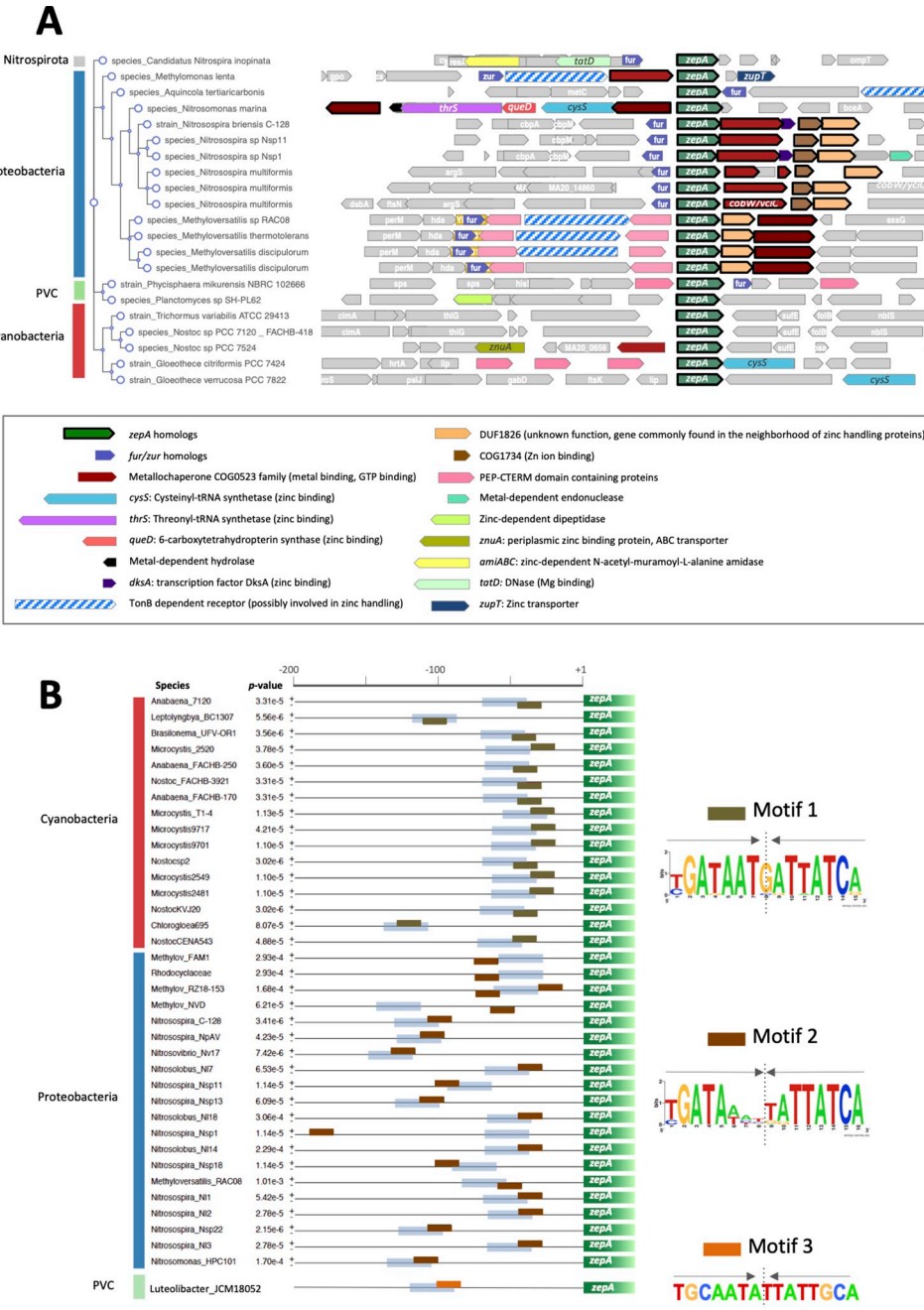

**Fig 7. Genomic evidences of the relationship of ZepA with zinc and Zur.** (A) The genomic region surrounding *zepA* in genomes from Nitrospirota, Cyanobacteria, Planctomycetota, and Proteobacteria was analyzed with GeCoViz software. Conserved genes around *zepA* identified by GeCoViz are shown in colors and depicted as arrows with a thick lane. Colored arrows with a thin line indicate other conserved genes identified by the authors, genes encoding proteins related to the handling of zinc, or other metals and genes encoding proteins with a PEP-CTERM domain. All other genes are in gray color. The identity of colored genes or their products and their relation with zinc or other metals is indicated at the bottom. (B) Palindromic motifs enriched in the 200 bp upstream region of *zepA* genes from species of Nitrospirota, Cyanobacteria, Proteobacteria, and the PVC superphylum were analyzed with MEME software. The position of enriched motifs 1 and 2 is shown as brown boxes and putative promoter sequences predicted with BPROM are shown as blue bars. The consensus sequence of motifs 1 and 2 was obtained by computing with Weblogo software. The sequences identified by MEME are shown on the right side. A vertical line indicates the symmetry axis and arrows are depicted to indicate the palindromic nature of these sequences.

In bacteria, regulatory genes governing small regulons are commonly proximal to their targets [43,44]. The proximity of *zepA* homologs to putative *zur* genes prompted us to check for the existence of ZBSs in the upstream regions of *zepA* genes. Since the ZBS sequence may be slightly divergent in distinct species, we performed a blind search for palindromic sequences enriched in such regions (Fig 7B and S1 File). Sequences matched 2 highly similar motifs, a 15 bp motif (motif 1), with matching sequences upstream of cyanobacterial *zepA* genes, and a 16 bp motif (motif 2) with matching sequences upstream of *zepA* genes from Proteobacteria. Both motifs were highly similar to ZBSs reported for a variety of species (see e.g., [5]), motif 1 perfectly matching the consensus site reported for *Anabaena* [25]. Interestingly, in most cases, both motifs overlapped or were just downstream of predicted promoter sequences (blue bars in Fig 7B). Although not found by computer search, a perfect palindromic sequence (motif 3) highly similar to the ZBS reported for Clostridia was observed upstream of *zepA* in *Luteolibacter yonseiensis* (PVC superphylum) (Fig 7B). This sequence also overlapped a predicted promoter (Fig 6B). As a whole, observations of sequence similarity, structure conservation, gene neighborhood, and putative Zur sites supported the involvement of ZepA homologs in zinc handling.

Experimental evidence on this was sought by setting up an assay in the ZepA-tolerant strain Lemo21(DE3) of *E. coli*. This assay was based on the observation that IPTG-induced expression of *Anabaena* ZepA provoked lethality in minimal medium in the presence of EDTA, growth being rescued by the specific addition of zinc (Fig 8A, sector 5), but not other metals, to the medium (Fig 8B, sector 5). This observation suggested that *Anabaena* ZepA interfered with zinc uptake, perhaps by sequestering external zinc that does not reach the cytoplasm due to the probable absence of downstream partners for ZepA in this host, causing lethality. Crucially, ZepA homologs from Proteobacteria, Nitrospirota, or the PVC superphylum behaved in this assay similarly as *Anabaena* ZepA (Fig 8A and 8B, sectors 3, 4, and 6), evidencing their specificity for zinc and reinforcing the idea that ZepA homologs play a similar role in zinc handling as *Anabaena* ZepA.

## Phylogeny of ZepA

The patchy distribution of *zepA* among species from distant phyla (Fig 5A) and its high degree of sequence conservation suggested that it could have been transmitted by horizontal gene transfer (HGT) between species rather than being passed down vertically through evolution. However, phylogenetic evidence indicated the opposite, i.e., when the *zepA* gene tree and 16S species tree were compared, no significant differences were found (AU test *p*-value = 0.5). This indicated that the vertical transmission of the gene was largely dominant in its evolution. Hence, *zepA* appears to have descended vertically from the common ancestor of each phylum with rare, if any, contribution of HGT between phyla (Figs 9 and S9). This implies that the *zepA* gene has a very ancient origin, which has important implications for our understanding of its function and evolution.

## Discussion

Many studies have characterized the role of cytoplasmic, plasma membrane, periplasmic, and outer membrane proteins in zinc handling [5] but data on extracellular proteins have been lacking so far. In this work, we provide evidence showing that in *Anabaena* Zur modulates the composition of the exoproteome, and we show that one exoprotein controlled by Zur mediates zinc supply to the cell, providing a first insight into the role of this protein and its homologs.

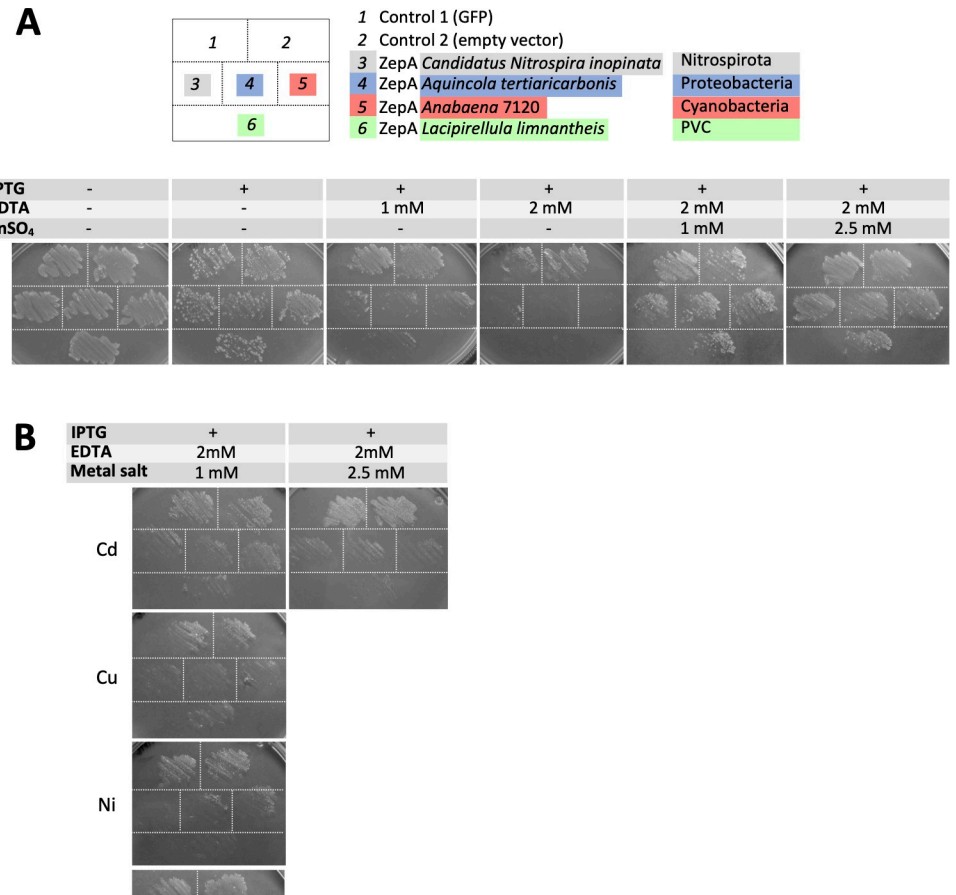

**Fig 8. Functional test of *Anabaena* ZepA and ZepA homologs in *E. coli*.** (A) Growth test of Lemo21(DE3) *E. coli* strain expressing the indicated proteins from the pET28b vector. Control 1 expresses GFP and Control 2 contains the empty pET28b plasmid. The diagram at the top indicates the position of each strain in the plates at the bottom that contained M9 medium with the indicated supplements. (B) M9 plates were supplemented with cadmium acetate, copper sulphate, nickel sulphate, or cobalt chloride at the indicated concentration. Plates containing 2.5 mM of any of the 3 latter salts showed no growth of any of the strains and are not shown. Strains were the same as in (A) and were plated following the same distribution as in (A). Growth tests in (A) and (B) were repeated 3 times with identical results.

## Role of Zur in the composition of the *Anabaena* exoproteome

The basis for the reduced abundance of U-proteins in the exoproteome of the Δ*zur* mutant may be manifold. One possibility is that Zur fails to activate some of these genes in the mutant. Consistently, putative ZBSs have been identified in the promoter of the *furA* gene (*all1691*) (S3 Table), which is in line with previous observations of Zur binding to the upstream sequence of this gene [26]. Of note, ZBSs localize far upstream from the TSS, which might be compatible with activation. This finding reinforces the proposed cross-talk between Zur and FurA regulators in *Anabaena* [26]. Accumulation of Zur-regulated O-proteins (i.e., ZepA, ZnuA, Alr3242, and Alr4028-4029) would be mainly due to de-repression in the Δ*zur* mutant. However, we do not rule out that side effects may influence on the abundance of O-proteins. For example, similar to *zur* mutants from a variety of species, the absence of Zur would provoke a permanent

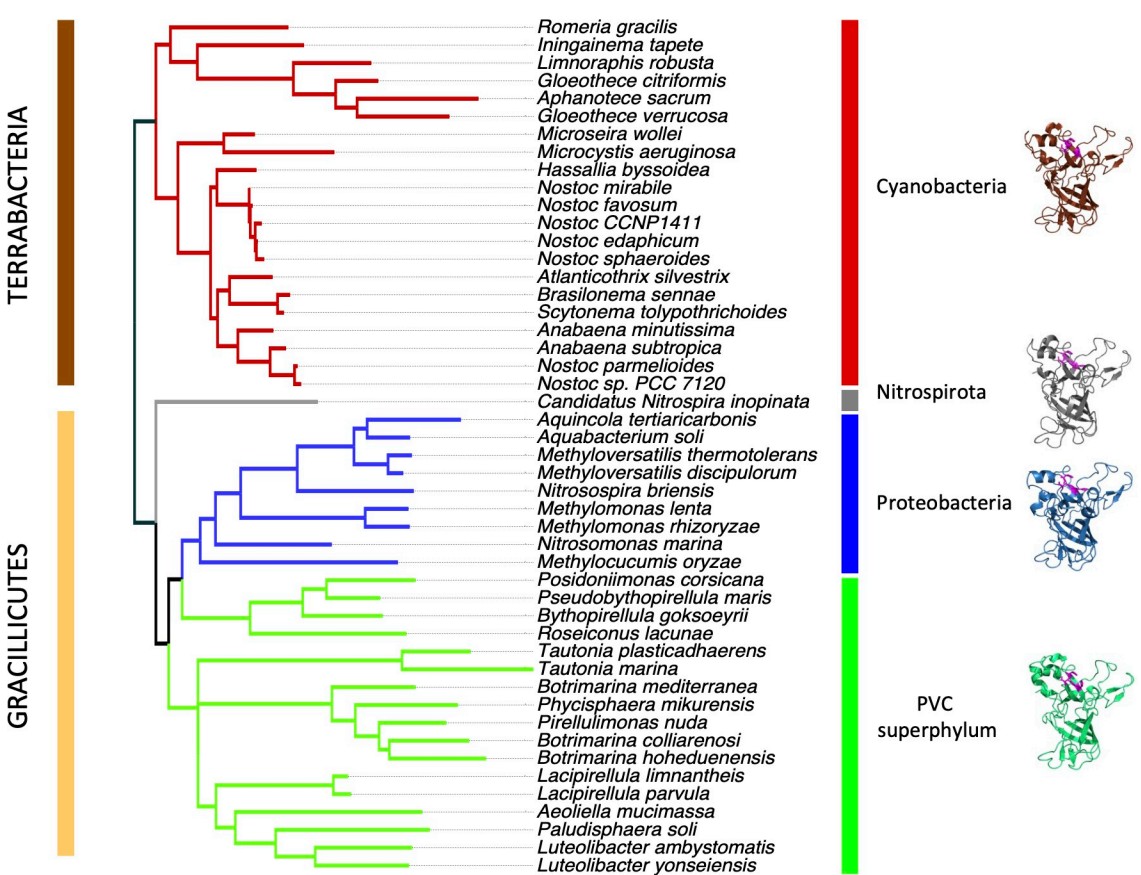

**Fig 9. Phylogenetic tree based on the sequence of ZepA proteins.** The names of the corresponding species are indicated. The data underlying this figure can be found in S1 Data. The predicted model structure of a ZepA representative for each group is shown on the right side. Putative zinc-binding histidine residues are depicted in purple color.

induction of zinc uptake systems and a high level of internal zinc in the *Anabaena* Δ*zur* strain [45], which would impinge on regulators and systems for handling other metals [24,46,47] and would alter the cytoplasmic redox state [26,48,49]. Side effects associated with this altered status may indirectly influence on the accumulation of proteins in the extracellular space.

## Function of O- and U-proteins

U-proteins' function is related to regular cell functioning under normal conditions and it is difficult to envisage the functional basis for their low abundance in the Δ*zur* mutant. The 4 O-proteins regulated by Zur are peripheral proteins likely involved in the supply of zinc to the cell. For example, ZnuA (All0833) is the substrate-binding protein of a zinc-specific transport system of the plasma membrane. Alr3242 and Alr4028-4029 are TBDTs and are the only ones among the 22 TBDTs encoded in the *Anabaena* genome [50] under the control of Zur [25]. TBDTs are outer membrane (OM) proteins that mediate energy-dependent transport of substrates like bacteriocins, metalophores, vitamin B-12, or carbohydrates from the extracellular space to the periplasm [51,52]. Genes encoding TBDTs are also part of the Zur regulon in a variety of gram-negative bacteria (see [5] and references therein), including *Caulobacter crescentus*, where they were shown to be important under zinc limitation conditions [53]. Notably, ZnuA and TBDTs are known to be operative in the periplasm and the OM, respectively. Determining whether the presence of ZnuA, Alr3242, and Alr4028-4029 in the extracellular space is

a consequence of their easy release due to their peripheral localization or they somehow contribute to zinc handling in this compartment is important.

ZepA is also a direct target of Zur and results in this work point to a role in the supply of zinc to the cell (Figs 1C, 2B, and 2C). Evidence indicates that ZepA is a genuine exoprotein (Fig 1C) with N- and C-terminal signal sequences apparently cleaved upon exportation (Figs 4A, 4B, and S2D). The role of N-terminal signal peptides in protein translocation has been well characterized [54]. By contrast, information on the PEP-CTERM domain is scarce. This domain has been proposed to be excised by EpsH exosortase proteins when the PEP motif is located at the outer layer of the plasma membrane. After proteolysis, the protein is supposed to be subjected to further modification and sorting steps to be targeted to or through the outer membrane [34,55], which is consistent with the identification of ZepA in the exoproteome of *Anabaena* in this work (Figs 1C and 4A). Of the 2 subfamilies of EpsH exosortases in cyanobacteria [55], a single representative (All0497) of the B subfamily is found in *Anabaena*, which is probably responsible for the processing of ZepA. Interestingly, *Anabaena* ZepA is also exported when expressed in *E. coli*, indicating that this host counts with the necessary compatible machinery for the recognition and processing of the ZepA exportation signals, as expected from the existence of genes encoding proteins with a signal peptide and a PEP-CTERM domain in its genome.

It is worth stressing that ZepA is constitutively expressed (Fig 2B) and operates under normal conditions (Fig 3A), which makes it part of the housekeeping machinery for zinc acquisition in *Anabaena*. Besides, ZepA is also part of the acclimation system to zinc deficiency, increasing in abundance in these conditions (Figs 1C and 2B). The mechanism by which ZepA delivers zinc to the cell is unknown. A plausible hypothesis is that it transfers the metal to an OM receptor that transports it into the periplasm (S3 Fig), though the import of the zinc-ZepA complex as a whole into the periplasm cannot be ruled out, as the import of large proteins through TBDTs has been described [56]. However, we do not favor this second possibility as ZepA was not detected in the periplasm of *Anabaena* (Fig 1C, lane 2, notice that cell extracts of the Δ*zur* strain contain periplasmic material), and because import into the periplasm would pose difficulties for the recycling of ZepA molecules and could provoke competition for zinc with ZnuA. Whichever the case, interaction of ZepA with an outer membrane receptor appears a plausible hypothesis. Very interestingly, it was recently reported that azurin, an extracellular protein of *Pseudomonas aeruginosa* involved in the import of copper, interacts with an outer membrane TBDT named OprC [57]. It would be interesting to investigate whether ZepA interacts with any of the 2 TBDTs induced by zinc limitation in *Anabaena*. The requirement for an OM receptor would be compatible with the lethal phenotype induced by ZepA and its homologs in *E. coli*. Zinc sequestration by ZepA and the probable absence of a specific receptor in this host being the cause of lethality.

## Role of extracellular components in zinc uptake

The requirement of an OM receptor for a diffusible protein as ZepA would have an obvious advantage in nature, i.e., ZepA would lock zinc away from competitors that do not express the receptor. However, a secreted molecule that diffuses away in the extracellular space could not find its way back, bringing no benefit to the producer cell. Mathematical models developed for siderophores indicate that these diffusible molecules are most effective in physically structured habitats (i.e., microbial mats, biofilms, or soil) where diffusion is limited [58]. In these densely populated environments, the diffusible molecule can benefit the producer cell or clonemates that display the receptor. We propose that ZepA functions in a similar way. Interestingly, it has been reported that bacteria that produce proteins with a PEP-CTERM domain are most

frequently associated with sediments, soil, and biofilms [34]. The use of ZepA as a zinc scavenging system also appears in line with the strong competition for nutrients in such structured habitats.

The basis for the identification of ZepA in liquid culture in our study is likely a consequence of the approach used. ZepA was identified in the exoproteome of a Δ*zur* strain, where the absence of Zur provokes constitutive expression and secretion of ZepA, regardless of the medium (solid or liquid) or condition utilized. By contrast, in a wild-type strain, ZepA would be expressed only under zinc limitation conditions, which are expectedly more common in structured environments. Consequently, in natural settings, ZepA would be expressed and secreted primarily in habitats where this protein is anticipated to be most effective for zinc acquisition.

Transport of substances into the cell is commonly mediated by membrane-embedded transport systems. However, it has been known for decades that extracellular molecules like siderophores also contribute to the import of iron [58]. More recently, siderophore-like molecules have been shown to mediate the import of other metals, and are termed metalophores, or zincophores when specific for zinc. Gene clusters involved in zincophore synthesis have been described in several bacteria and evidence was obtained on the involvement of TBDTs in the import of zinc-zincophore complexes [14–16,59]. Results in this work bring into evidence that not only zincophores but also extracellular proteins like ZepA can contribute to the import of zinc. Zincophores and ZepA appear as alternative, though perhaps non-mutually exclusive, strategies for scavenging zinc from the extracellular medium.

## ZepA origin and evolution

Multiple lines of evidence in this work (Figs 5–9) including (i) the high degree of sequence conservation; (ii) the similarity of modeled structures; (iii) the conservation of N- and C-terminal exportation signals; (iv) the conservation of residues predicted to bind zinc and other molecules; (v) the location of *zepA* homologs in zinc-related gene clusters; (vi) the presence of putative ZBSs upstream of *zepA* genes; and (vii) experimental results in this work strongly indicated that ZepA homologs from distant lineages are actual orthologs with a role in extracellular zinc handling.

Apparently, *zepA* has been transmitted mainly through vertical inheritance in 4 distant bacterial groups. Presumably, its toxicity in heterologous hosts, as observed in *E. coli*, may have hampered its transmission by HGT. Crucially, these groups belong to the 2 major lineages directly emerging from the LBCA, namely Terrabacteria (Cyanobacteria) and Gracillicutes (Nitrospirota, Proteobacteria, and the PVC superphylum) [60]. This suggests that *zepA* was likely present in the common ancestor of these phyla, which is the LBCA. The alternative scenario of *zepA* being originally present in one particular group and transferred to others by early HGT events cannot be ruled out, but whichever the case, these data point to a very early origin of *zepA* in evolution, which has manifold implications. For example, the apparent conservation of the ZepA function in each lineage indicates that this role was established early. This is interesting because the conditions of the ancient Earth differed from those today; indeed, the availability of metals has changed through geological time [61–63]. Evidence indicates that the Great Oxygenation Event (GOE) that occurred 2.4 to 2.1 billion years ago as a consequence of the biological activity of cyanobacteria provoked a dramatic reduction in iron availability and a rise in zinc and copper. Zinc is thought to have been particularly scarce in the anaerobic Archaean environments before the GOE, its availability rising from the Proterozoic to the present day [63]. Our results indicate a very early origin for ZepA, which may predate the GOE (notice that cyanobacteria necessarily evolved previously). Hence, this protein

may have been operating as a system for zinc acquisition in ancient bacteria dwelling in environments with a poor content of this metal. Consistent with this hypothesis is the observation that in *Anabaena*, ZepA is induced when zinc is scarce and favors survival under these conditions (Fig 3). Massive loss of *zepA* along bacterial evolution is also consistent with the rise in zinc availability during the Proterozoic, which would have turned this system unnecessary, facilitating its loss or its replacement by other systems. However, its remarkable conservation for billions of years and its functionality in present-day bacteria evidences the usefulness of *zepA* for organisms that may encounter conditions reproducing the scarcity of zinc in Archaean ecosystems.

### The Zur-*zepA* pair along evolution

The proximity of *zepA* to putative *zur* genes and the presence of putative ZBS in *zepA* promoters of distant groups suggest that the Zur-*zepA* regulatory pair may have been operating from the early days of each lineage, which would entail that Zur is also an early evolving protein. Most regulators of the FUR family are responsive to metals, having diverged from a single founding member of unknown specificity. It has been reported that the evolvement of metal-coordinating protein folds has run in parallel to the availability of metals through geological time [61–63], and it is reasonable to think that the speciation of FUR regulators would have likewise be determined by environmental conditions that selected for each new species. Results in this work allow us to propose that Zur may have been one of the first regulators of the FUR family to evolve, consistently with the scarcity of zinc previous to the GOE, whereas the iron-specific Fur regulator would have evolved at a later stage when the rise of oxygen turned iron into a limiting metal for bacteria.

Numerous studies increasingly focus on the extracellular space as a place of intense biological activity, not only as the source of nutrients and the sink for cellular waste but also as a space where cells locate small molecules, proteins, and membrane-coated vesicles to interact with the environment and with other organisms. Results in this work have helped to identify several proteins of the exoproteome of *Anabaena* that change in abundance when the Zur regulator is inactivated and when zinc is limiting. Evidence indicates that one of these proteins, ZepA, constitutes a zinc uptake system operating in bacteria from the early days of evolution to the present.

## Materials and methods

(See extended Materials and methods in S2 File)

### Organisms and growth conditions

*Anabaena* sp. strain PCC 7120 was cultured in shaken Erlenmeyer flasks with BG11 medium supplemented with 10 mM $NaHCO_3$ or in glass cylinders aerated with a mixture of air enriched with 1% $CO_2$ containing the same medium [64] at 28 to 30˚C, under white light illumination (30 μmol of photons $m^{-2}$ $s^{-1}$) provided by Osram LED 16.4 W/4000K lamps. Solid medium was prepared by the addition of agar at 1% final concentration to BG11. Cultures were supplemented with N,N,N′,N′-tetrakis (2-pyridilmethyl) ethylenediamine (TPEN) when indicated. TPEN was prepared in dimethyl sulfoxide (DMSO). In these DMSO was added to control cultures (not containing TPEN) at the same final concentration as those with TPEN. Generation of the Δ*zur* mutant was described in [25]. Δ*zur* mutant cultures were supplemented with 2 to 5 μg·$ml^{-1}$ streptomycin and 2 to 5 μg·$ml^{-1}$ spectinomycin. Cultures of *all3515* mutants were supplemented with 25 μg·$ml^{-1}$ neomycin. Functional tests shown in Fig 3 were

performed by spotting serial dilutions of cell suspensions containing 1 μg·ml$^{-1}$ of chlorophyll on BG11 plates containing the indicated supplements.

*Escherichia coli* DH5α was routinely used for cloning. *E. coli* was grown in Luria–Bertani [65] medium supplemented with antibiotics when needed at the following concentrations: ampicillin, 50 μg·ml$^{-1}$; kanamycin, 25 μg·ml$^{-1}$; chloramphenicol, 25 μg·ml$^{-1}$; streptomycin, 25 μg·ml$^{-1}$; spectinomycin, 100 μg·ml$^{-1}$. Growth tests in *E. coli* Lemo21(DE3) shown in Fig 8 were performed in solid M9 minimal medium containing 42 mM $Na_2HPO_4$, 22 mM $KH_2PO_4$, 8.5 mM NaCl, 18 mM $NH_4Cl$, 1 mM $MgSO_4$, 0.1 mM $CaCl_2$, 0.1% (w/v) glucose, and 1.5% (w/v) agar.

## Generation of *zepA/all3515* mutants

A 0.7 Kb DNA fragment internal to the *zepA/all3515* gene was amplified with primers ALL3515-2F and ALL3515-2R and cloned between the BamHI and XhoI sites of the pRL278 vector, which does not replicate in *Anabaena*. The resulting pCMA21 plasmid was introduced in *Anabaena* by conjugation as described [66]. Ex-conjugants that have integrated the plasmid by single recombination were selected in plates containing 25 μg·ml$^{-1}$ neomycin and tested for segregation using primers ALL3515-3F and PRL278-1F to amplify the mutant allele or primers ALL3515-3F and ALL3515-3R to amplify the wild-type allele. Segregated ex-conjugants lacking the wild-type allele were selected for further work. Oligonucleotides used in this work are listed in S4 Table.

## Growth assay of wild-type *Anabaena* and *all3515* mutants on solid media and statistical analysis

Approximately 5 μl of cell suspensions containing 1 μg of chlorophyll ml$^{-1}$ and 2-fold serial dilutions were spotted on plates of BG11 medium with the indicated supplements (Fig 3) and cultured for 15 days under standard conditions. Plates were photographed 8 days after plating. Each experiment was repeated 3 times. Growth was quantified with ImageJ software by analyzing the cell density in each spot. In order to this, images were converted to binary and the density within circles enclosing each spot was quantified. For statistical analysis, a Growth Index (GI) was defined as the sum of the density of all spots of one strain in a plate. This index is indicative of the capacity of each strain to grow in a particular medium. The statistical significance of the differences in growth in a particular medium was calculated by comparing the GI of one of the mutants versus the wild type using the Student's *t* test.

## Preparation of exoproteome samples for mass spectrometry identification

A total of 80 ml cultures of wild-type *Anabaena* and the *Δzur* mutant cells were set up in duplicate in Erlenmeyer flasks at an optical density at 750 nm ($OD_{750}$) of 0.15. Cells were obtained by gentle centrifugation at room temperature of precultures grown to late exponential phase and suspensions were prepared in fresh BG11 medium supplemented with 8.8 mM $NaHCO_3$. Cultures were incubated for 12 days 28°C under illumination (30 μmol of photons m$^{-2}$ s$^{-1}$) with agitation. After this period, cultures were in the late exponential phase of culture ($OD750$ = 1-1.3). Cultures were sequentially filtered through 0.45 and 0.22 μm filters MCE filters (Filter-Lab). The flow through fraction was immediately frozen at −80°C and lyophilized. Lyophilized material was resuspended in 2.5 ml of buffer A (50 mM Tris-HCl (pH 7.5) and Complete EDTA free protease inhibitor cocktail) and dialyzed against 3 L of buffer A, performing 2 overnight steps. After dialysis, the sample was transferred to 15 ml tubes, frozen at −80°C, and lyophilized. Samples were resuspended in 800 μl of buffer containing 50 mM Tris-HCl (pH 7.5), 1 mM PMSF, 1 mM EDTA, 2% SDS, and a cocktail of protease inhibitors and

were centrifuged at 20,000 $g$ for 2 min to eliminate particles. The supernatants were subjected to precipitation with trichloroacetic acid (TCA)/acetone.

## Protein digestion and mass spectrometry analysis

The comparative quantitative proteomic analysis of the exoproteomes of *Anabaena* wild type and *Δzur* mutant was performed using 2 biological replicates from each strain, each sample analyzed in triplicate. The precipitated protein extracts were resuspended in 20 μl of 50 mM ammonium bicarbonate supplemented with 0.2% RapiGest (Waters) and total protein concentration was quantified using Qubit system (Invitrogen). Approximately 10 μg of protein were incubated with 4.5 mM dithiothreitol (DTT) for 30 min at 60°C followed by incubation with 10 mM chloroacetamide for 30 min in darkness at room temperature. Trypsin treatment was performed at 37°C at a 1:40 ratio (Trypsin:Protein). Digestion was stopped by the addition of formic acid. A mixture of SCIEX synthetic peptides was also added to each sample to a final concentration of 50 fmol/μl for posterior normalization of chromatograms necessary for SWATH processing. Mass spectrometry analysis was carried out at the Proteomics Service of the Instituto de Bioquímica Vegetal y Fotosíntesis (Seville, Spain). The analyses were performed in a triple quadrupole-time of flight (Q-TOF) hybrid mass spectrometer (5600 plus, Sciex), equipped with a nano electrospray source coupled to an Eksigent model 425 nanoHPLC. Analyst TF 1.7 was the software used to control the equipment, as well as for the acquisition and processing of data. Peptides were first loaded onto a trap column (Acclaim PepMap 100 C18, 5 μm, 100 Å, 100 μm id × 20 mm, Thermo Fisher Scientific) isocratically in 0.1% formic acid/2% acetonitrile (v/v), at a flow rate of 3 μl/min for 10 min. Subsequently, they were eluted on a reverse phase analytical column (Acclaim PepMap 100 C18, 3 μm, 100 Å, 75 μm id × 150 mm, Thermo Fisher Scientific) coupled to a PicoTip emitter (F360-20-10-N- 20_C12, New Objective). Peptides were eluted over a linear gradient of 2% to 35% (v/v) of solvent B in 60 min at a flow rate of 300 nL/min. As solvents A and B, formic acid 0.1% (v/v) and acetonitrile with formic acid 0.1% (v/v) were used, respectively. The source voltage was selected at 2,600 V and the heater temperature remained at 100°C. Gas 1 was selected at 15 psi, gas 2 to zero, and the curtain to 25 psi. Initially, a spectral library was built for SWATH analysis, by data-dependent acquisition (DDA) method, consisting of a TOF-MS Scan between 400 and 1,250 m/z, accumulation time of 250 ms, followed by 50 MS/MS (230 to 1,500 m/z), accumulation time of 65 ms and with a total 3.54 s cycle time. Protein identification was performed using the ProteinPilot software (version 5.0.1, Sciex) with the Paragon Algorithm. Data obtained was search against the UniProt proteome *Nostoc* sp. PCC 7120 FASTA (downloaded on 15/07/2021, entries: 6,070) combined with the SCIEX contaminants database. Automatically generated reports in ProteinPilot were manually inspected for FDR; cutoff proteins with only proteins identified at an FDR ≤1% were considered for subsequent analyses. The obtained library recorded 210495 MS/MS spectra and 51943 MS/MS spectra were assigned to 4,690 peptide sequences (1% FDR). SWATH-MS analysis was performed using 3 technical replicates of 2 biological replicates using a DIA method. Each sample (1 μg of protein) was analyzed using the SWATH-MS acquisition method consisting of a repeated acquisition cycles of TOF MS/MS scans (230 to 1,500 m/z, 60 ms acquisition time) of 60 overlapping sequential precursor isolation windows of variable width (1 m/z overlap) covering the 400 to 1,250 m/z mass range from a previous TOF MS scan (400 to 1,250 m/z, 50 ms acquisition time) for each cycle with 3.68 s total cycle time. Autocalibration of the equipment and chromatographic conditions were controlled by an injection of a standard PepCalMix (Sciex) between replicates. SWATH quantification was performed with PeakView 2.2 software (Sciex) with the MicroApp SWATH 2.0 and MarkerView software (version 1.2.1.1, AB Sciex). Spectral alignment was performed

with PeakView using the spectral library built from the DDA runs (1%FDR). Afterwards, the processed files containing protein information were loaded into MarkerView for normalization of protein intensity (total area sums). SWATH quantification was carried out with 9,722 spectra assigned to 1,397 peptides sequences.

## Electrophoresis mobility shift assays (EMSA)

Oligonucleotides used for the generation of probes are listed in S4 Table. A DNA probe of the upstream region of *zepA*/*all3515* was generated by annealing partially overlapping oligonucleotides ALL4725-GS-1F and ALL3515-R and filling in with Klenow DNA polymerase (Thermo Scientific) in the presence of [$^{32}$P]-dCTP. Control DNA probes for the positive and negative controls were generated by annealing partially overlapping oligonucleotides ALL4725-GS-1F and ALL4725-GS-1R (positive control) or ALL4725-GS-1F and ALL4725-GS-2R (negative control) and filling in with Klenow. A total of 20 μl reactions contained 0.1 to 0.5 fmol DNA in a buffer containing 20 mM Tris-HCl (pH 8), 50 mM KCl, 1 mM dithiothreitol (DTT), 20% glycerol, and 0 to 10 pmol of protein. Reactions were incubated for 15 min at room temperature and resolved on a 5% native acrylamide gel.

## RNA extraction and northern assays

RNA preparation from cyanobacterial cells and northern assays were carried out as described in [67] with modifications. Briefly, cells from 80 ml cultures were harvested by filtration through nitrocellulose 0.45 μm pore size filters (Millipore), washed with 50 mM Tris (pH 7.5), 100 mM EDTA and resuspended in 100 μl of resuspension buffer (300 mM glucose, 10 mM sodium acetate (pH 4.5)). The suspension supplemented with 400 μl of lysis buffer (2% SDS, 10 mM sodium acetate (pH 4.5)) and 100 μl of EDTA (pH 8). This mixture was extracted once with 1 ml of phenol at 65°C, once with 1 ml of phenol:chloroform (1:1) and once with 1 ml of chloroform. RNA was precipitated by the addition of 1 ml of isopropanol and washed with 70% ethanol. The pellet of nucleic acids was resuspended in 90 μl of RNase-free water and incubated in the presence of Turbo DNase (Ambion) for 1 h at 37°C. The enzyme was subsequently inactivated by addition of Dnase Inactivation Reagent (Ambion). For northern blot assays, 5 to 10 μg of RNA was resolved in a formaldehyde-containing 1.2% agarose gel, transferred to Genescreen plus nylon membranes (Perkin Elmer), and hybridized to a $^{32}$P-labeled probe specific for the *zepA*/*all3515* gene, which was generated by PCR with primers ALL3515-3F and ALL3515-3R (S4 Table). For normalization, the same membranes were hybridized with a probe specific for the *rnpB* gene generated with primers A7120-RNPB-1F and A7120-RNPB-1R (S4 Table).

## Bioinformatics

Prediction of subcellular localization was performed with the PSORTb software version 3.0.3 [68] available at https://www.psort.org/psortb/. Prediction of signal peptides was made with SignalP-6.0 software [32] available at https://services.healthtech.dtu.dk/service.php?SignalP. Prediction of nonclassical protein secretion was carried out with Secretome P 2.0 (with the recommended cutoff score for bacteria (0.5) and the type of secretion system was predicted using BastionX V2.0 [69] at the Bastion Server page (https://bastionhub.erc.monash.edu/bastionxPrediction.jsp). Putative ZBSs upstream of genes encoding O- or U-proteins were identified by scanning 500 bp regions upstream of the translational start site using the FIMO program [70] available at the MEME Suite (https://meme-suite.org/meme/). Scanning for palindromic sequences enriched in the upstream regions of *zepA* homologs was performed using the MEME program [71] of the MEME suite. BPROM was used for bacterial protein

prediction [72]. SMART [73] was used to identify protein domain architectures and GeCoViz [74] for neighborhood analysis.

ZepA biological diversity was explored by searching for its homologs in the Refseq_select curated database. Refseq_select contains a representative set of reference and representative genomes, reducing bias due to overrepresented strains [75] and visualizing inter-species protein diversity. In Refseq_select assemblies generated from environmental samples are excluded due to concerns with the accuracy of the organism assignment and possible cross-contamination. The number of prokaryotic assemblies at Refseq_select (updated on 19th August 2022) was 16157 (15626 Bacteria) according to NCBI Boolean search using the query (prokaryota [orgn] AND ("representative genome"[refseq category] OR "reference genome"[refseq category]). All searches were done using BLASTP 2.13.0+ [76,77] with default settings. All3515 (ZepA) homologs were mapped into bacterial phylogeny and taxonomy using AnnoTree [78] and the BLASTP XLM2 output file input. This web server uses up-to-date phylogeny and taxonomy from GTDB [79].

For the ZepA phylogeny, All3515 homologs retrieved from the NCBI Refseq_select database as of August 2022 and were aligned using the L-INS-i strategy of MAFFT v7 [79] and trimmed with BMGE 1.1 [76] using the default setting for relaxed trimming. ZepA phylogenetic reconstruction was performed using the W-IQ-TREE web server [77] and the best available model (WAG+I+4G+F) according to Modelfinder [80]. We used AU-test [81] as implemented in W-IQ-TREE to estimate the *P*-value of the species tree topology to be in the confidence set of ZepA trees. The species tree was based on an alignment of 16S sequences and its corresponding optimal evolutionary model (GTR+4R) according to the PhyML model selection [82]. Small ribosomal RNA sequences were retrieved from RNAcentral and aligned considering secondary structure with SINA v1.2.11 as implemented in the SILVA database [83].

## Structure modeling

AlphaFold2 was used to predict the mature protein structures of ZepA through the use of ColabFold [84]. We used the default setting (MMseqs2+Uniref+Environmental, Unpaired and paired) for all predictions except for that of *Anabaena* sp. PCC 7120 in which a multiple sequence alignment (MSA) was used. To construct this MSA, we retrieved 230 full-length homologs from the nr-NCBI database using a BLASTP 2.13.0 search with the mature sequence of All3515/ZepA as the query. These sequences were then aligned using the MAFFT v7 (L-INS-i algorithm). This procedure slightly increases the accuracy of ZepA structure prediction.

To predict ligand binding and generate the resulting protein–ligand structure, we utilized the online resources MIB2 [38] and Galaxysite [85] with the AlphaFold2 predictions of the ZepA structures. It should be noted that both ligand identification procedures require the availability of experimentally resolved protein structures of protein–ligand complexes. A caveat for potential biases in these analyses due to unbalanced representation of reference proteins in the databases must be mentioned (i.e., whereas metalloproteins are abundant in the Protein Data Bank, crystallized proteins bound to the great diversity of microbial molecules as hopanoids or glycolipids are scarce). The structure of ZepA was compared to structures in the databases using the DALI server [86] (http://ekhidna2.biocenter.helsinki.fi/dali/).

## Statistical analysis

The R package limma v3.54.1 [87] was used to assess the differential expression of exoproteins. Briefly, normalized data were log2-transformed, median centered and the 3 technical replicates averaged before fitting the data to a linear model (lmFit, robust method), and using moderated

t-statistics with Limma's empirical Bayes method to analyze differential expression. The $p$-values were adjusted for multiple hypotheses testing using Benjamini–Hochberg FDR correction. Proteins were classified as differentially expressed if the adjusted $p$-value (q-value) was less than 0.05.

## Supporting information

**S1 Fig. Reproducibility of biological replicates for exoproteome characterization.** Scatter plots showing the correlation between the normalized protein intensities (log2 transformed) of the 2 biological replicates of *Anabaena* sp. PCC 7120 (WT) (A) and Δ*zur* strain (B). Pearson's correlation coefficient ($R^2$) is shown. The data underlying this figure can be found in S1 Data. (PPTX)

**S2 Fig. Features of the All3515 protein and generation of *all3515* mutants.** (A) Sequence of All3515/ZepA showing the N-terminal signal peptide and the C-terminal PEP-CTERM domain. Histidine residues predicted to coordinate a zinc atom are depicted in pink color, histidine residues of the N-terminus of the mature protein are shown in orange color and acidic residues in this region are in green. (B) Comparison of the C-terminal All3515 sequence to the archetypal PEP-CTERM domain described by Haft and colleagues. (C) A scheme indicating the steps followed for the construction of the *all3515* mutants is shown. Segregation analysis of the mutants is shown at the bottom of the figure. Vertical dashed lines on the gel image are included solely to indicate that irrelevant parts of the gels are not shown. (D) Western blot of cell fractions of *E. coli* Lemo21(DE3) pET28b:ZepA_Ana and the Δ*zur* mutant of *Anabaena*. Lanes 1–5 were loaded with fractions from *E. coli* Lemo21(DE3) pET28b:ZepA_Ana. Lane 1 was loaded with a cell extract (CE) corresponding to 50 μl of culture, lanes 2 and 3 were loaded with cell extract (CE) or spheroplast extract (Sphe) corresponding to 40 μl of culture, lane 4 was loaded with periplasmic fraction (Peripl) corresponding to 500 μl of culture, lane 5 was loaded with extracellular material (EM) corresponding to 1.25 ml of culture, lane 6 was loaded with extracellular material corresponding to 6 ml of culture of the Δ*zur* mutant of *Anabaena*. The gel of the top panel was subjected to western blot with a specific antibody against ZepA. Unspecific bands are labeled as "*u*." Bands reacting specifically to the anti-ZepA antibody are labeled as "*a*," "*b*," or "*c*" according to their size. A vertical dashed line is depicted to indicate that parts at each side of the line were exposed differently. The bottom panel is an identical gel stained with Coomassie. (PPTX)

**S3 Fig. Model for the function of ZepA.** The picture shows a theoretical model based on results presented in this manuscript of zinc acquisition mediated by ZepA in *Anabaena*. Zinc atoms are depicted as yellow dots. OM stands for outer membrane and PM for plasma membrane. Cylinders on the outer membrane represent Zur-regulated TBDTs All3242 and All4028-4029. Dashed arrows indicate steps that will need to be further investigated. (PPTX)

**S4 Fig. Sequence conservation and coverage analysis, predicted accuracy of models, and prediction of metal-binding residues of ZepA from *Anabaena* and *Lacipirellula limnantheis*.** (A) Coverage and identity analysis of the multiple sequence alignment provided to Alphafold2 for structure prediction. Two juxtaposed plots are shown in this panel. One plot shows the coverage and identity of sequence homologs represented as complete or interrupted horizontal lines ordered from less (bottom) to more (top) conserved with respect to the mature sequence of ZepA of *Anabaena* (the query). Each line is depicted according to the color scale of the sequence identity shown on the right side. Gaps in the alignment are shown as empty

spaces. The second plot in this left panel corresponds to the black line, which represents the number of sequences (left vertical axis) in which a given position of the query protein is conserved. The right panel is similar but uses the ZepA protein from *Lacipirellula limnantheis* (PVC superphylum) as the query protein. In this case, homologous sequences to the query protein were selected by Alphafold2 (mmseqs2_uniref_env setting). (B) Predicted lDDT per position on a scale from 0–100 of 5 AlphaFold2 models colored by rank. The average pLDDT of the most confident prediction (rank1) is 93.5 (for *Anabaena*) and 95.2 (for *Lacipirellula*). Model confidence varies significantly along the chain, but most positions are modeled with high accuracy (pLDDT > 90) and can serve to characterize binding sites. In the 5 models, accuracy decreased between 30 and 70 at the 10 N-terminal amino acids (pLDDT < 50) predicting a disordered region (unstructured under physiological conditions). The yellow inset represents the output of the MIB2 metal binding prediction server showing metal binding scores for each amino acid (derived from sequence and structure conservation measures) as a yellow line. Scores higher than 2 predict a metal-binding residue (red circle). Vertical gray bands were depicted to show that predicted metal binding sites are located at regions where the model confidence is lower, which also coincides with regions of low sequence conservation in the plot in (A). Histidines 44, 53, and 55 of the *Anabaena* ZepA protein are indicated. (C) Confidence in the relative position and orientation of different model parts as Predicted Aligned Errors (PAE). Blue colors indicate confidence in the relative position of domains. (D) PDB summary (PDBsum) of the predicted structural models. Residues belonging to domain A are highlighted in orange, and those of domain B are in cyan. Histidines of the putative zinc-binding pocket are labeled with a dot. (E) Close view of the putative zinc-binding pocket of the ZepA protein of *Anabaena* (left) and *Lacipirellula* (right) showing the distances between N atoms in histidine residues and the zinc atom in the model. The data underlying this figure can be found in S1 Data.
(PPTX)

**S5 Fig. Model structures of ZepA from species of distinct bacterial phyla.** (A) Structural models for the ZepA proteins from the indicated organisms were built using AlphaFold2 and are shown. (B) Structural models were aligned with PyMOL v 1.7.6.3 and are shown in different orientations. (C) Close view of the *Anabaena* ZepA model showing residues putatively involved in zinc coordination.
(PPTX)

**S6 Fig. Prediction of metal-binding residues and conservation of the spatial arrangement of putative zinc-binding histidines in ZepA homologs.** (A) Plots correspond to the output of the MIB2 metal binding prediction server. The *y* axis indicates the MIB2 score and the *x* axis indicates the amino acid positions. The species and the phylum of each ZepA homolog is indicated. Histidines predicted to form the zinc-binding pocket are enclosed in a frame and indicated with a purple bar. Notice that in *Gloeothece verrucosa*, only 2 of the 3 histidine residues are conserved. The consensus sequence of the putative zinc-binding pocket is shown at the top with histidines residues in purple. The data underlying this figure can be found in S1 Data. (B) Pictures show a close view of the zinc-binding pocket of each ZepA homolog. The structure of each protein was modeled with AlphaFold2. Putative zinc-binding histidine residues are depicted in purple color.
(PPTX)

**S7 Fig. Docking sites in the ZepA proteins from *Anabaena* sp. PCC 7120 and *Lacipirellula limnantheis*.** The structure of these proteins predicted by AlphaFold2 was submitted to the Galaxysite server (https://openebench.bsc.es/tool/galaxysite). Molecules predicted to interact

with these proteins are shown together with the localization of the docking site. We interpret that the planar lipid, would likely correspond to a hopanoid instead of a planar sterol as predicted by the Galaxysite server. Planar sterols are rare in bacteria but are highly similar to bacterial hopanoids. Notice that the 2 structures are viewed from distinct positions.
(PPTX)

**S8 Fig. Conserved interaction sites of ZepA from species of distinct bacterial phyla.** Structural models obtained with AlphaFold2 for the ZepA protein of the indicated species were used to interrogate the MIB server or the GalaxySite server to determine putative interaction sites. Conserved sites are shown as solid boxed and absent sites as empty boxes. Zn, zinc; LP4, a lipopolisacharide (2-deoxy-3-O-[(3R)-3-hydroxytetradecanoyl]-2-{[(3R)-3-hydroxytetrade-canoyl]amino}-4-O-phosphono-beta-D-glucopyranose); C3S, a planar sterol (cholest-5-en-3-yl hydrogen sulfate); NAG, N-acetyl glucosamine (2-acetamido-2-deoxy-beta-D-glucopyranose); MAN (alpha-D-mannopyranose); OLA (oleic acid).
(PPTX)

**S9 Fig. Comparison of phylogenetic trees based on the sequence of ZepA or on the sequence of 16S rRNA.** Species names are connected to their corresponding branch by solid or dashed gray lines. As shown, the position of phyla in both trees mirrored each other, which is indicative of rare horizontal transfer events between phyla. The data underlying this figure can be found in S1 Data.
(PPTX)

**S10 Fig. Quantitation and statistical analysis of the growth of wild-type *Anabaena* and *all3515* mutants in solid media.** Plates in Fig 3 and 2 biological replicates of this experiment were photographed and growth in each spot was quantitated using ImageJ software as indicated in Materials and methods. A Growth Index (GI) was defined as the sum of the density of all spots of one particular strain in a plate. This index is indicative of the capacity of each strain to grow in a particular medium. Colors in the heatmaps correspond to the average GI for each strain and condition according to the scale shown on the left. Darker color indicates a higher growth capacity. For statistical analysis, the average GI of a mutant strain in one condition was compared to the average GI of the wild type in the same condition using the Student's $t$ test. * Indicates $p < 0.05$, ** indicates $p < 0.01$, not significant differences are not indicated. The data underlying this figure can be found in S2 Data.
(PPTX)

**S1 File. Palindromic sequences enriched in the upstream region of *zepA* genes.** Palindromic sequences found enriched in the regions below by the MEME program (https://meme-suite.org/meme/tools/meme) are highlighted in yellow. The palindromic sequence in the upstream region of the *zepA* gene from *Luteolibacter yonseiensis* is highlighted in green. Promoter sequences (−35 and −10 boxes) predicted by BPROM (http://www.softberry.com/berry.phtml?topic=bprom&group=programs&subgroup=gfindb) are shown in red. The last nucleotide of each sequence corresponds to the −1 position.
(DOCX)

**S2 File. Extended Materials and methods.**
(DOCX)

**S1 Table. Comparison of the set of proteins identified in this work with previously reported exoproteomes of *Anabaena*.** The set of proteins identified and quantified by SWATH as components of the exoproteome of *Anabaena* sp. PCC 7120 and the Δ*zur* mutant with the previously published exoproteomes of *Anabaena* sp. PCC 7120 from Oliveira and

colleagues and Hahn and colleagues in BG11 medium. Venn diagram performed with Venny 2.0 available in https://bioinfogp.cnb.csic.es/tools/venny/index2.0.2.html.
(XLSX)

**S2 Table. Results of limma statistical analysis on the data obtained from SWATH-MS.** Additional data (Uniprot ID, Locus tag, location, COG and computational prediction for the presence of signal peptide, protein secretion and subcellular localization) is shown for differentially expressed proteins. The table presents the estimation of Log2-fold-change [log2FC($\Delta zur$/WT)] for the contrast of WT and $\Delta zur$ exoproteomes. The left (CI.L) and right (CI.R) limits of the 95% confidence interval for log2FC provide a measure of the precision of the log2-fold change estimation. The table also includes the average log2-expression (AveExpr), the moderated t-statistic (t), the *p*-value, and the false discovery rate (FDR or q-value) obtained after Benjamini–Hochberg correction of the *p*-value. Additionally, the table includes the log-odds measure (B), which indicates the likelihood that the protein is differentially expressed. Prediction of subcellular localization, of signal peptides, of nonclassical protein secretion, and type of secretion systems was performed as indicated in Materials and methods.
(XLSX)

**S3 Table. Zur binding sites in the 5′ upstream region of genes encoding O- and U-proteins.** The position of putative Zur-binding sites respect to transcription start sites (TSSs) is depicted at the right side. "TSS not mapped" indicates that no TSS has been reported at a distance less than 500 bp from the Zur-binding site.
(DOCX)

**S4 Table. Synthetic DNA used in this work.**
(DOCX)

**S1 Data. Numerical data for Figs 1A, 1B, 4C, 4D, 4E, 5A, 6A, 9, S1A, S1B, S4A, S4B, S4C, S6A, and S9.**
(XLSX)

**S2 Data. Numerical data for S10 Fig.**
(XLSX)

**S1 Raw Images. Original images of blots and gel results.**
(PDF)

## Acknowledgments

We are indebted to Rocío Rodríguez from the Proteomics Facility of IBVF for excellent assistance with sample preparation for mass spectrometry analyses and the analysis of proteomics data. We are grateful to Dr. M. Esther Pérez-Pérez (IBVF, Sevilla) for advise on western blots.

## Author Contributions

**Conceptualization:** Jesús A. G. Ochoa de Alda, María F. Fillat, Ignacio Luque.

**Investigation:** Cristina Sarasa-Buisan, Jesús A. G. Ochoa de Alda, Cristina Velázquez-Suárez, Miguel Ángel Rubio, María F. Fillat, Ignacio Luque.

**Methodology:** Cristina Sarasa-Buisan, Jesús A. G. Ochoa de Alda, Cristina Velázquez-Suárez, Miguel Ángel Rubio, Guadalupe Gómez-Baena, Ignacio Luque.

**Supervision:** Jesús A. G. Ochoa de Alda, María F. Fillat, Ignacio Luque.

**Writing – original draft:** Ignacio Luque.

**Writing – review & editing:** Cristina Sarasa-Buisan, Jesús A. G. Ochoa de Alda, Cristina Velázquez-Suárez, Miguel Ángel Rubio, Guadalupe Gómez-Baena, María F. Fillat, Ignacio Luque.

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
