## [Editor Report · Decision Letter 0]

2 May 2023

Dear Dr. Luque, 

Thank you for submitting your manuscript entitled "Characterization of a cyanobacterial exoproteome reveals a novel zinc acquisition system used by primitive bacteria" for consideration as a Research Article by PLOS Biology.

Your manuscript has now been evaluated by the PLOS Biology editorial staff, as well as by an academic editor with relevant expertise, and I am writing to let you know that we would like to send your submission out for external peer review.

Once your full submission is complete, your paper will undergo a series of checks in preparation for peer review. After your manuscript has passed the checks it will be sent out for review. To provide the metadata for your submission, please Login to Editorial Manager (https://www.editorialmanager.com/pbiology) within two working days, i.e. by May 04 2023 11:59PM.

Kind regards,

Paula

---

Senior Editor

PLOS Biology

---

## [Decision Letter · Decision Letter 1]

7 Jun 2023

Dear Dr Luque,

Thank you for your patience while your manuscript entitled "Characterization of a cyanobacterial exoproteome reveals a novel zinc acquisition system used by primitive bacteria" was peer-reviewed at PLOS Biology. It has now been evaluated by the PLOS Biology editors, an Academic Editor with relevant expertise, and by three independent reviewers. 

The reviews are attached below. As you will see, the reviewers find the conclusions of the manuscript interesting, but they also raise several concerns that would need to be addressed before we can consider the paper for publication. While Reviewer 2 is mostly satisfied, Reviewers 1 and 3 suggest several experiments and clarifications to strengthen the results.

After discussing the reviews with the Academic Editor, we have decided to invite you to submit a revision that thoroughly address the reviewers' reports. Given the extent of revision needed, we cannot make a decision about publication until we have seen the revised manuscript and your response to the reviewers' comments. Your revised manuscript is likely to be sent for further evaluation by all or a subset of the reviewers.

**IMPORTANT - SUBMITTING YOUR REVISION**

3. Resubmission Checklist

a) *PLOS Data Policy*

b) *Published Peer Review*

Sincerely,

Ines

--

Ines Alvarez-Garcia, PhD

Senior Editor

PLOS Biology

on behalf of

Senior Editor

PLOS Biology

Reviewers' comments

Reviewer #1:

The manuscript proposed by Sarasa-Buisan and co-authors is a good piece of research that characterizes the Zur-controlled exoproteome and its role in zinc acquisition by bacteria. Unfortunately, several shortcomings have been identified and clarifications are needed. 

Major concerns:

1) The authors should exploit the shotgun proteomics data in order to search for the exact coverage of the ZurA protein and establish the exact maturation sites (N-terminal signal peptide processing and C-terminal PEP-Cterm domain excision). Alternatively, they could analyze the global molecular weight of the proteins (LC-MS) and establish the exact limits of the ZurA polypeptide. 

2) The authors did not demonstrate what is the quaternary structure of ZurA and whether ZurA binds zinc, and did not confirm the number of zinc atoms per protein. This would be an important result to complement their study, easily obtained after heterologous expression of the zurA gene. 

3) In the results section, the description of the proteomic experiment (lines 127-135) is not so clear. What was the efforts in terms of mass spectrometry? The number of MS/MS spectra recorded and the number of MS/MS spectra assigned to peptide sequences? How many biological replicates have been documented? What was the reproducibility of the analysis on these replicates? The authors mentioned in the M&M section the use of DDA and DIA analysis. The results section is unclear regarding these two strategies. Why were both strategies needed? Are the outcomes of the DDA and the DIA consistent?

4) Here, why was the cellular proteome not analyzed at the same time? It would be interesting to highlight the proteins more specifically detected in the exoproteome compared to the cellular proteome with an enrichment factor. This would allow to highlight proteins that could come from cell lysis.

5) Figure 2 shows that the strategy for analysing vesicle protein content is different from the exoproteome analysis. The authors should have performed shotgun proteomics also on these vesicles in order to gain more insights. The rationale of this experimental change is difficult to understand.

Minor concerns

6) The abstract needs rephrasing to improve its interest and be more exact. The abstract should gain clarity with the following rephrasing: "Bacteria have developed fine-tuned responses to cope with potential zinc limitation. The Zur protein is a key player in coordinating this response in most species. Comparative proteomics conducted on the cyanobacterium Anabaena highlighted the more abundant proteins in a zur mutant compared to the wild type. Experimental evidence showed that the exoprotein ZepA mediates zinc uptake. Genomic context of the zepA gene and protein structure prediction prvided additional insights. Phylogenetic analysis suggests that ZepA represents a primordial system for zinc acquisition that has been conserved for billions of years in a handful of species from distant bacterial lineages. Furthermore, these results show that Zur may have been one of the first regulators of the FUR family to evolve, consistently with the scarcity of zinc in the ecosystems of the Archean eon.".

7) In the introduction, the sentence "The external medium also contains proteins that constitute the so-called exoproteome" requires a general reference because the audience may not be familiar with the meaning of the exoproteome.

8) The sentence "Zur has been well characterized in a few species. Remarkably, in the oceanic cyanobacterium Synechococcus sp. WH8102, Zur exhibits unique features regarding the orientation of the two domains and the regulatory site, which is distinct and involves coordination residues different from those in non cyanobacterial Zur." needs also a reference. 

9) Line 153: replace ORF by CDS if you are talking about polypeptide sequence. 

10) Line 167: please indicate the fold change threshold.

Reviewer #2:

Sarasa-Buisan et al. present data that identify exoproteome proteins that show changes in abundance, up or down, in a zur mutant strain. These data are used to identify a protein they have named ZepA that is involved in zinc handling. They show that ZepA is regulated by Zur and identify Zur binding sites close upstream of the TSS1 and TSS2. Growth experiments show that ZepA is required for growth when zinc is removed by chelation and protects against high zinc levels. Bioinformatics supports a ZepA relationship with zinc handling and phylogenetics indicated that ZepA may have evolved very early for zinc acquisition and that Zur may be one of the first members of the Fur family. This paper is thorough and presents a complete story; it is well organized and is well written. I found no major flaws in the experimental design, data, or presentation in the manuscript and have only a few minor comments for the authors to consider in revising their manuscript.

The following comments should be considered by the authors. 

L 48. Replace "consistantly" with "consistant".

L 255, Fig. 3C. The authors should comment on the effect of TPEN in the zur mutant. Maybe I missed it, but is there a hypothesis for the reduced level of transcripts in the presence of TPEN?

L 302. "Zur-dependent induction" is confusing because Zur is a repressor. This should be rephrased.

L 357. The red arrows in Fig.6 are too small.

L 372. "lane" should be "line".

L 602. "obeyed to" is unclear here. This needs to be rephrased.

L 664. The degree symbol is an "a" here.

Supplementary Fig. S1 legend. Fix "estrategy"

Supplementary Fig. S2 legend. Fix "confronted"; and "seldom" should probably be "rare".

Reviewer #3:

In this manuscript the authors describe a study which uses an exoproteomic profiling comparing wild type Anabaena and a zur mutant to characterize extracellular zinc response. They also identified proteins in extracellular vesicles. They find a novel exoprotein overexpressed in the mutant, which they functionally characterize with EMSA to show Zur binding, Northern blots to show overexpression in the Zur mutant, and knockout mutant phenotyping. They then conduct bioinformatic analyses to find homologs of the new protein, genomic context of the homologs, phylogenetic analyses, and structural modeling. They conclude that the new protein ZepA mediates Zn uptake and interacts with the Zur regulator and that it is an ancient protein that was present in the LCBA and was lost in most lineages. 

This manuscript represents an impressive amount of work and has some interesting and valuable results in characterizing a new putative Zn homeostasis protein. However I have some major concerns:

1. It currently feels like 3-4 different stories (exoproteome, extracellular vesicles, ZepA function in Anabaena, ZepA evolution), based on only a few actual experiments which feel somewhat incomplete and/or incompletely described. I think the experimental results most clearly suggest a focus on the functional role of ZepA in Anabaena—all the other pieces are distracting and periphery, with much less support from the data. This is most clear in the Discussion, which needs to be narrowed and focused. E.g. the results presented do not justify an entire page on how exo-proteins reach the extracellular space, since the experiments were not designed to test this—there was 1 experiment on exoproteins that did not compare to intracellular proteomes and only ran duplicate biological replicates at one time point under one condition. The experiment compared the wild type and zur mutant so that is the data that should be the focus.

2. The experimental results are intriguing, but do not go far enough to fully explain and characterize ZepA function—ZepA is supposedly extracellular, and thus you would expect a very different phenotype on solid vs liquid media, but the limited phenotyping of the mutant was done on solid medium, and not quantified statistically for significance. This feels like a missed opportunity, which ideally could be rectified, but at the very least should be noted. Further, a schematic model showing the proposed mode of action/regulation of ZepA in Anabaena Zn homeostasis should be added to aid interpretation of the results and what the unknowns still are. And possibly also compared to Zn homeostasis in another organism that has lost ZepA.

3. ZepA evolution—this feels like a tangent, and while interesting, is not supported by the experimental results presented. The experimental work is done in one strain, and then the bioinformatic work extrapolates the same function to all the other homologs (e.g. line 575 "the apparent conservation of the ZepA function")—I think this needs to be better constrained in the interpretation since the function of the other homologs was not tested experimentally, nor are there experimental results in the literature to support the function of ZepA in the other taxa. Yes, the genomic context and structural modeling suggests the ZepA homologs also play a role in Zn homeostasis, this is still just a hypothesis until there is some experimental results. To then take the known homologs and discuss the evolution of this novel protein throughout the bacterial tree of life is too far without characterizing at least one other member.

More detailed comments:

Exoproteomes:

1. Need to address cell lysis as a possible origin for the exoproteins, especially given the culturing to late exponential. Best would be to compare to intracellular proteomes, although this might not exist for the zur mutant.

2. Also need to be explicit that it is duplicate biological replicates (triplicates are usually required for statistical analyses), but at the very least make it clear in figure legends and results, and I think some of the "quantitative comparison" statements should probably be re-phrased.

3. I also have some concern that the mutant may have different cell lysis rates than the wild type, so you could expect more extracellular proteins in the mutant than the wild type, or that if the mutant and wild type grew differently and were only sampled at one time point then you would see more intracellular proteins in one strain vs the other—a time course could have helped with this, but at the very least provide details on Zn levels and growth rates of both strains for that experiment.

4. I'm not sure I'm convinced that ZepA only functions extracellularly—since they didn't compare to intracellular proteomes, it could be more abundant inside the cells—at the very least they could look in the literature for proteomics to see if it is detected intracellularly as well?

5. All the information on the exoproteins aside from ZepA are not presented in a clear fashion, all the TSS start site data (Lines 166-191) is very difficult to read through. Some more concise description would help, integrating with the next sections on ZepA.

6. Methods: 

a. why continuous light? 

b. How did the authors ensure late exponential growth and not stationary? 

c. What was the Zn concentration in the media in the exoproteome experiment?

d. Why dialyzed and then precipitated? Was this similar to the protocol compared to in the other papers cited?

Extracellular Vesicles:

This part is not integrated well at all with the rest of the paper, and not completely described in the methods or results—I would suggest removing it entirely. The method for extraction appears to be general for bacteria (although it was not detailed at all and merely references the paper)—was there any verification of whether it was successful or biased? As I understand vesicle extraction is not straightforward between different taxa and extraction may need to be modified.

ZepA function:

EMSA: Isn't there usually a negative control too, not just a positive?

For the RNA and Northern extracts, methods are sparse—one publication referenced, and not in that organism. Also no information on replicates or any brief description of the methods.

No description of the methods for phenotyping the mutants—were replicates run? That data is not quantitative thus difficult to know whether interpretation is correct. No statistics on that analysis either. 

Genome context, phylogeny and structural modeling:

1. The genome context work is nice and does support and similar role for the homologs (although this still needs to be verified experimentally)

2. The phylogeny part is outside of my expertise, but with so few homologs the interpretation that the 16S and gene trees are similar does not seem that compelling to mean that vertical transmission is likely, we may just be missing key taxa.

3. The structural modeling is also nice, as is the prediction of metal binding, and provides some good hypotheses to test experimentally. The figures and text could both be made more concise.

4. All of these pieces could be made more concise and constrained better to the experimental results and literature to make a more clear story.

---

## [Decision Letter · Decision Letter 2]

1 Feb 2024

Dear Dr Luque,

Thank you for your patience while we considered your revised manuscript "Characterization of a cyanobacterial exoproteome reveals a novel zinc acquisition system used by primitive bacteria" for publication as a Research Article at PLOS Biology. This revised version of your manuscript has been evaluated by the PLOS Biology editors, the Academic Editor, and two of the original reviewers.

Based on the reviews, we are likely to accept this manuscript for publication, provided you satisfactorily address the remaining points raised by reviewer #3, and the following data and other policy-related requests.

IMPORTANT - please attend to the following:

a) Please change your Title to: "A widespread and ancient bacterial zinc acquisition system identified from a cyanobacterial exoproteome" - this highlights the discovery over the approach.

b) Please attend to the remaining requests from reviewer #3. I discussed these with the Academic Editor, who said "I do not believe it is critical to have the metagenomic search completed for this manuscript. Depending on their bioinformatics expertise, this might not be as trivial as the reviewer indicates, and I would hate for this great manuscript to be held up any longer." We will therefore leave this requirement optional.

c) Please address my Data Policy requests below; specifically, we need you to supply the numerical values underlying Figs 1AB, 4CDE, 5A (treefile), 6A, 9 (treefile), S1AB, S4ABC, S6A, S9 (treefile), either as a supplementary data file or as a permanent DOI’d deposition.

d) Please cite the location of the data clearly in all relevant main and supplementary Figure legends, e.g. “The data underlying this Figure can be found in S1 Data” or “The data underlying this Figure can be found in https://doi.org/10.5281/zenodo.XXXXX”

e) Please make any custom code available, either as a supplementary file or as part of your data deposition.

We expect to receive your revised manuscript within two weeks. 

*Published Peer Review History*

*Press*

Sincerely,

Roli Roberts

Roland Roberts, PhD

Senior Editor

rroberts@plos.org

PLOS Biology

DATA POLICY:

Regardless of the method selected, please ensure that you provide the individual numerical values that underlie the summary data displayed in the following figure panels as they are essential for readers to assess your analysis and to reproduce it: Figs 1AB, 4CDE, 5A (treefile), 6A, 9 (treefile), S1AB, S4ABC, S6A, S9 (treefile). NOTE: the numerical data provided should include all replicates AND the way in which the plotted mean and errors were derived (it should not present only the mean/average values).

CODE POLICY

Per journal policy, as the code that you have generated is important to support the conclusions of your manuscript, we require that you make it available without restrictions upon publication. Please ensure that the code is sufficiently well documented and reusable, and that your Data Statement in the Editorial Manager submission system accurately describes where your code can be found.

We require the original, uncropped and minimally adjusted images supporting all blot and gel results reported in an article's figures or Supporting Information files. We will require these files before a manuscript can be accepted so please prepare and upload them now. Please carefully read our guidelines for how to prepare and upload this data: https://journals.plos.org/plosbiology/s/figures#loc-blot-and-gel-reporting-requirements

DATA NOT SHOWN?

REVIEWERS' COMMENTS:

Reviewer #1:

The authors answered appropriately to my previous concerns and improved greatly their manuscript. The supplementary material is of high quality and much appreciated. Congratulations for this research. 

Reviewer #3:

[identifies herself as Rhona K. Stuart]

The manuscript is much improved, and most of my concerns have been addressed. 

One that I don't think was fully addressed is regarding the phenotyping shown in Figure 3—where is the data for the other times it was run? It's not enough to just say it was replicated three times. At the very least put it all in supplemental so that the reader doesn't have to take it on faith. But even better, instead of pictures all that data could be averaged and quantified and some stats tests for significance applied. 

Also, I made a point in my previous review about the solid vs liquid media comparisons, and still think there should be some mention of this in results/discussion—the discussion talks about structured environments retaining ZepA—Anabaena certainly expresses proteins and exoproteins differently in liquid versus plate growth, so these are not really equivalent. It's interesting that you see a phenotype you expect in solid medium, but it should be noted in result or discussion this is a different cultivation environment than the exoproteomics experiment. 

I think the ZepA evolution has been better described and qualified. I think it would be even more convincing to look at genomes from metagenomics datasets, or just look for ZepA occurrence in those datasets. For the hypothesis that only "structured environments" would retain ZepA, you wouldn't expect to see it in aquatic datasets but yes in mats or soils. However, this is more of a suggestion, (which would help convince me of what the authors argue regarding it's evolution).

Minor comments:

Line 46: "additional insights" is very vague. Consider specifying.

Line 49: "Consistently" to "consistent".

Line 86: "zinc traces" to "trace zinc"

---

## [Editor Report · Decision Letter 3]

12 Feb 2024

Dear Dr Luque,

Thank you for the submission of your revised Research Article "An ancient bacterial zinc acquisition system identified from a cyanobacterial exoproteome" for publication in PLOS Biology. On behalf of my colleagues and the Academic Editor, Emiley Eloe-Fadrosh, I'm pleased to say that we can in principle accept your manuscript for publication, provided you address any remaining formatting and reporting issues. These will be detailed in an email you should receive within 2-3 business days from our colleagues in the journal operations team; no action is required from you until then. Please note that we will not be able to formally accept your manuscript and schedule it for publication until you have completed any requested changes.

Sincerely, 

Roli Roberts

Senior Editor

PLOS Biology

rroberts@plos.org